# Learning plasma dynamics and robust rampdown trajectories with predict-first experiments at TCV

Allen M. Wang [1,2] ✉, Alessandro Pau [3], Cristina Rea[1], Oswin So[2], Charles Dawson[2], Olivier Sauter [3], Mark D. Boyer[4], Anna Vu[3], Cristian Galperti[3], Chuchu Fan[2], Antoine Merle[3], Yoeri Poels [3], Cristina Venturini[3], Federico Felici[3], Stefano Marchioni[3], TCV Team*

The rampdown phase of a tokamak pulse is difficult to simulate and often exacerbates multiple plasma instabilities. To reduce the risk of disrupting operations, we leverage advances in Scientific Machine Learning (SciML) to combine physics with data-driven models, developing a neural state-space model (NSSM) that predicts plasma dynamics during Tokamak à Configuration Variable (TCV) rampdowns. The NSSM efficiently learns dynamics from a modest dataset of 311 pulses with only five pulses in a reactor-relevant high-performance regime. The NSSM is parallelized across uncertainties, and reinforcement learning (RL) is applied to design trajectories that avoid instability limits. High-performance experiments at TCV show statistically significant improvements in relevant metrics. A predict-first experiment, increasing plasma current by 20% from baseline, demonstrates the NSSM's ability to make small extrapolations. The developed approach paves the way for designing tokamak controls with robustness to considerable uncertainty and demonstrates the relevance of SciML for fusion experiments.

Upcoming burning plasma tokamaks, such as SPARC[1] and ITER[2], require reliable plasma control to avoid operational delays and machine damage due to plasma disruptions, a challenge that will only increase for tokamak pilot plants[3], like ARC[4] and DEMO[5]. Given that this risk becomes intolerable at high plasma current, $I_p$, and stored energy, $W_{tot}$, a key mitigation strategy is to de-energize the plasma by performing a rampdown of the plasma current, but doing so typically pushes the plasma closer to multiple instability boundaries[6–8]. Figure 1 depicts the phases of a tokamak pulse, beginning with rampup of the plasma current to the steady-state flattop phase, and ending with a rampdown. Notably, Fig. 1 also shows an example of a quantity correlated with plasma instability growing during the rampdown phase, a challenge that motivates the algorithmic design of safe rampdown trajectories. This challenge is especially acute in reactor-relevant high-

performance (HP) plasmas, which operate close to instability boundaries to achieve the high normalized plasma density, typically quantified by the Greenwald fraction $f_{GW}$[9], and normalized plasma pressure, $\beta_N$[10], important for economical energy production. The importance of designing robust rampdowns for reactor-relevant fusion plasmas is highlighted by the recent record-breaking HP campaign at the Joint European Torus (JET), for which most disruptions occurred during the termination phase[11]. For the baseline scenario, a ≈ 15% increase of the plasma current, from 3MA to 3.5MA, increased the disruptivity considerably from ≈20% to ≈50%[11]. This challenge motivates the development of tools that can rapidly adapt rampdown trajectories to manage disruptivity as fusion performance is increased.

Due to the stochasticity of plasma dynamics, hardware and control imperfections, and the possibility of off-normal events (ONEs), it is

[1]Plasma Science and Fusion Center, Massachusetts Institute of Technology, Cambridge, MA, USA. [2]Laboratory for Information and Decision Systems, Massachusetts Institute of Technology, Cambridge, MA, USA. [3]Swiss Plasma Center (SPC), Ecole Polytechnique Fédérale de Lausanne (EPFL), Lausanne, Switzerland. [4]Commonwealth Fusion Systems, Devens, MA, USA. ✉e-mail: awang@psfc.mit.edu

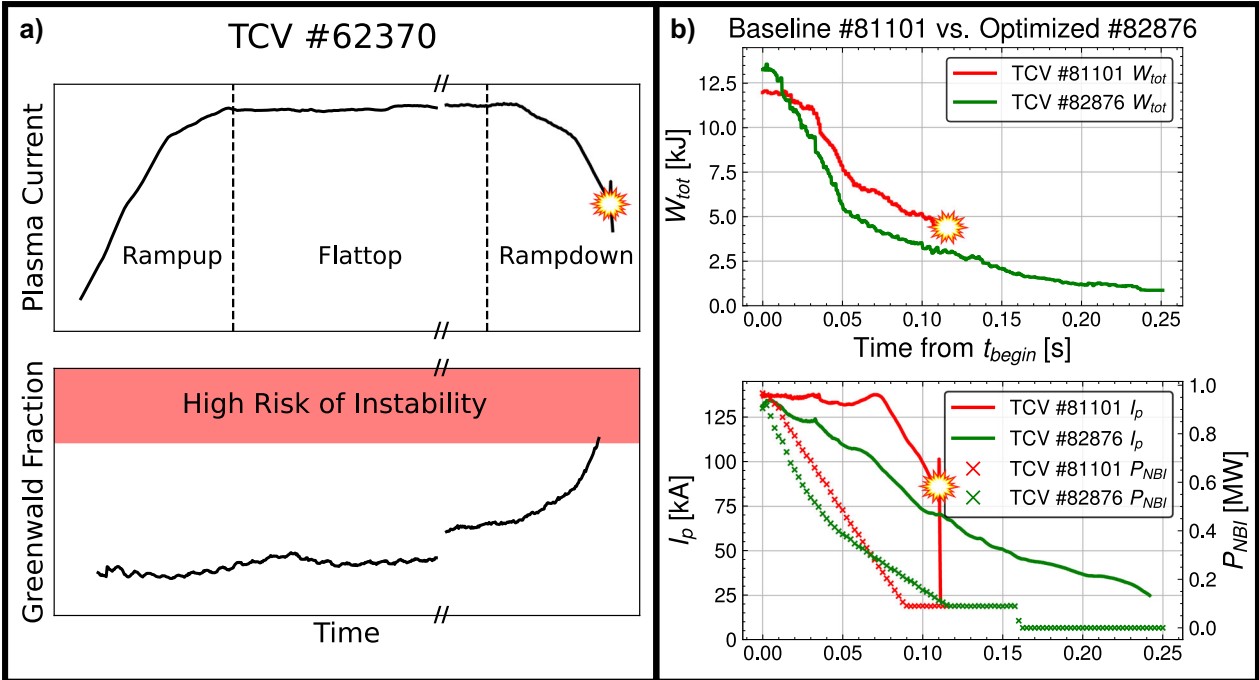

**Fig. 1 | Examples of plasma disruptions during rampdowns, and an example non-disruptive result obtained after deployment of the developed method.** **a** Data from an illustrative TCV pulse (#62370), showing the rampup, flattop, and rampdown phases, which are defined by the plasma current $I_p$. The bottom subplot shows how the rampdown pushes the plasma closer to an instability limit, in this case, the Greenwald density limit, defined by the Greenwald fraction $f_{GW} = 1$. Note this limit is approximate due to the, at present, incomplete physics understanding of the density limit[73,74]. Also note that the flattop phase is abbreviated here to more clearly highlight the rampup and rampdown. **b** A comparison of the current, $I_p$, stored energy, $W_{tot}$, and neutral beam injection power, $P_{NBI}$, trajectories for a baseline shot and an optimized shot, showing a faster and non-disruptive decrease in the plasma current and stored energy relative to the baseline.

important to design scenarios, trajectories, and controllers with robustness to distributional uncertainty in the dynamics of the plasma. The biggest barrier to designing for robustness in this context is the difficulty of simulating plasma dynamics during the highly transient rampdown phase, during which multiple physical quantities, many of which are not well-modeled with a principles-based approach, can change drastically. Due to this challenge, prior rampdown studies using existing simulators[6,7,12–15] typically make assumptions on important effects like the confinement regime transition, which is subject to significant uncertainty. These simulation limitations motivated recent experiments at DIII-D designing rampdown trajectories with black box Bayesian Optimization on three control variables, which achieved improvements in the plasma current at time of termination[8]. This experiment showed that relatively small adjustments can make an outsized impact; however, reported pulses, also known as shots, were all at low performance, and a predictive model-based approach is desired for upcoming tokamaks. These limitations motivate the development of models that efficiently learn difficult-to-simulate dynamics from experimental data, and which are massively parallelizable across uncertainties to enable robust model-based design of trajectories.

To address these challenges, this work takes a data-driven approach, leveraging recent advances made by the Scientific Machine Learning (SciML) community[16–18] and new machine learning frameworks, namely JAX[19], which enable the training of dynamics models that combine physics-based equations with data-driven models. A data-driven approach is not without precedent; aircraft flight control and simulation primarily utilize data-driven models of aerodynamics derived from flight test data in lieu of computational fluid dynamics[20,21], often with classical linear state-space models (SSMs)[22]. While prior works on learning plasma dynamics using unstructured neural networks required large datasets, often spanning thousands of shots[23–25], we gain sample efficiency by embedding physics structure into a neural state-space model (NSSM)[26,27]. This model was trained to generate sufficiently accurate predictions using a modest amount of data, with 311 rampdowns at low performance and only five shots with incomplete rampdowns in the relevant HP regime, with $\beta_N > 2$ and near the density limit. The model is capable of simulating $\approx 10^4$ rampdown trajectories per second on a single A100 GPU, enabling the usage of the NSSM in a reinforcement learning (RL) training environment.

The RL environment is massively parallelized to design trajectories with robustness to uncertainties, including the initial conditions of the plasma and its time-varying dynamics. We leverage the capabilities of RL for offline design of robust trajectories, which is more readily applicable to the safety-critical settings of upcoming tokamaks than RL for real-time control[28–31]. A similar approach has previously been demonstrated at KSTAR for designing feed-forward trajectories that reach target states[32,33]. After a small number of initial trial shots, the plasma reliably terminated at low plasma current and stored energy for five consecutive HP shots, with statistically significant improvements relative to baseline, although we encourage caution in interpreting the statistics of the result due to the small sample size. As a test of the viability of this approach for performing small extrapolations in an incremental HP campaign, which upcoming tokamaks like SPARC and ITER will undergo, we design a rampdown trajectory and perform a predict-first experiment by increasing the plasma current by 20%, from 140 to 170 kA, for a high $\beta_N$ plasma near the density limit, a scenario for which zero shots of rampdown data exist for TCV. In this extrapolation test, we a priori predict the dynamics of key quantities to within sufficient accuracy to successfully terminate the plasma on both attempts.

The paper is organized as follows. We begin with an overview of the experiment and report the achieved rampdown improvements, as measured by the key figures of merits of plasma current $I_p$ and stored energy $W_{tot}$ at the time of plasma termination. Then, an overview of the NSSM is provided along with medium-scale validation metrics demonstrating its predictive power. This is followed by an analysis of two shots in the experiment demonstrating the importance of

accounting for control errors in trajectory design for preventing a class of disruptions known as vertical displacement events (VDEs). Then, an analysis of 140 kA shots in the experiment shows how incremental re-training between run days resulted in rampdowns that are both faster and less disruptive. Results from the predict-first extrapolation test are reported, demonstrating the ability of the NSSM to make small extrapolations. Finally, we discuss future work and implications for upcoming tokamaks like SPARC and ITER.

## Results

### Experiment overview

The reported experiment was conducted as part of the 2024 TCV integrated control, HP experimental campaign. Flattop plasmas operated at a HP of $\beta_N > 2.0$ and near the density limit with a highly elongated diverted geometry with $\kappa \approx 1.6$ and $q_{95} \approx 4$. Initial shots in the experiment operated at $I_p = 140$ kA, henceforth known as the baseline HP scenario, with a final extrapolation test at $I_p = 170$ kA. Successful rampdowns from these scenarios require careful management of multiple plasma instability limits that can be exacerbated by details of the plasma trajectories. At present, a comprehensive understanding of disruptive limits remains an open problem, motivating many works on machine-learning-based prediction of disruptions[34–38]. However, prior rampdown studies[7,8,12], a survey of disruption causes at TCV[39], and the present physics-based understanding of this high-density scenario motivated constraints on four plasma parameters. Namely, we impose constraints on the Greenwald fraction, $f_{GW}$, the vertical instability growth rate $\gamma_{vgr}$, the plasma poloidal beta $\beta_p$, and the edge safety factor $q_{95}$. Managing the density limit, which is correlated with the Greenwald fraction, in this scenario is a particular challenge for fast terminations, as the relatively long particle confinement timescale is a major constraint on the speed of the rampdown. The considered constraints are further discussed in the context of the reward function in the "Methods".

To address the problem, an NSSM dynamics model was trained on a modest dataset of past rampdowns, which contains only 5 incomplete rampdowns in the relevant HP parameter space, as shown in Fig. 2a, b. This NSSM is then used in an RL environment to optimize a reward function, designed to minimize time to a goal plasma current of 40 kA and stored energy of 0.5kJ without disrupting, as shown in Fig. 2c. The action space was chosen to be plasma current, $I_p$, shaping parameters $\kappa$ and $a_{minor}$ and neutral beam injection (NBI) power $P_{NBI}$. User-specified constraints were set on the Greenwald fraction $f_{GW}$, safety factor $q_{95}$, and vertical instability growth rate $\gamma_{vgr}$ as calculated with the method in ref. [40], and poloidal beta $\beta_p$. The optimized action trajectories were then manually programmed into the TCV plasma control system (PCS). The details of the reward function, chosen limits, and PCS programming process are further discussed in the "Methods".

Statistical significance of control results in fusion is typically difficult to establish due to the scarcity of experimental time and relevant data points. This experiment also faces this challenge, given that the ramp-down experiment involved only nine shots, two of which were dedicated to debugging a legacy software issue, with only five rampdowns in the database near the relevant HP regime with $\beta_N > 2$. We use these five shots as our control set and define two test sets: one with the debugging shots and one without. As shown in Fig. 2d, the Mann-Whitney U test[41] shows a statistically significant improvement in $W_{tot}$ ($p < 0.05$) at the time of plasma termination of the experimental rampdowns for both definitions of the test set. Improvements in $I_p$ are only statistically significant when we do not include the debugging shots. While the results of this statistical test are encouraging, we urge caution in its interpretation, given the small sample sizes involved, and the fact that tokamaks are highly drifting distributions in practice, with uncontrolled variables such as wall conditioning making a meaningful impact on experimental outcomes.

Every shot involved in the experiment is shown in Fig. 3, showing improvements in $I_p$ and $W_{tot}$ at the time of plasma termination over the course of the experimental runs. The unoptimized baseline rampdown

trajectory for this scenario was disrupted at relatively high current and stored energy in #81101 and #81102 at $I_p \approx 80$ kA and $W_{tot} \approx 4$ kJ. The experiment proceeded iteratively, with re-training of the NSSM on new data and trajectories done after shots #81635, #81745, #81751, and #81830. A preliminary optimized trajectory was deployed in TCV #81635, which reached the goal $I_p$ and $W_{tot}$ before disrupting, but post-shot analysis showed poor radial control and tracking of the target shape, which was determined to be due to a legacy software issue detailed in Fig. S1 in the Supplementary Information. Shots #81741 and #81745 were spent resolving this issue, with it properly resolved in #81751, as shown in Fig. S1. #81751 is still disrupted due to a VDE, which was found to be due to a large sensitivity of $\gamma_{vgr}$ to small control errors in the inner gap. After #81751, an uncertainty distribution on gap errors was added to the RL training environment to gain robustness to this uncertainty, with subsequent shots experiencing similar control errors but without similar increases in $\gamma_{vgr}$, demonstrating the importance of designing trajectories with robustness to real-world uncertainties, as further discussed in the section.

For the final run-day, trajectories were re-optimized, and predictions of the plasma dynamics were generated a priori for both two reprisals of the baseline HP scenario, but also for the extrapolation test. All four shots for both scenarios terminated successfully below the goal $I_p$, with the baseline scenarios realizing both faster and non-disruptive trajectories relative to baseline and successful a priori predictions of plasma dynamics for both scenarios.

### Medium-scale validation of NSSM predictions

The NSSM was developed and trained to predict the time-dependent dynamics of the set of observations in response to the set of actions listed in Table 1. The primary goal of the model is to predict the dynamics of key quantities relevant to completing the control task of a fast disruption-free plasma rampdown in response to actuation of controllable variables, to allow the trajectory optimization algorithm to decide on actions that avoid user-specified limits on key quantities correlated with disruptions.

The model underwent two training phases: an initial training phase on a wider dataset with 311 shots in the training dataset and 131 shots in the validation dataset. To improve the predictive power of the model for the relevant scenario, we fine-tune just the confinement scaling of the model by training only on 44 shots with $I_p \leq 200$ kA, with all other model weights frozen. Due to the relatively small size of the fine-tuning dataset, we did not separate out a validation or testing dataset for this fine-tuning phase. As shown in Fig. 4, the trained model is able to predict the time-dependent dynamics of key 0D kinetic and disruptive quantities to within tens of percent for full rampdowns in the validation dataset, even in the 95th percentile of error. The percent errors for $\gamma_{vgr}$ can be relatively large, but, as shown in Fig. S3 in the Supplementary Information, this is largely attributable to the small value of $\gamma_{vgr}$ of limited plasmas, as the absolute error is relatively low.

The NSSM was initially developed with a neural network predictor for the kinetic profiles on the full $\rho$ grid, and initial training runs found that the profile predictor can accurately predict kinetic profiles given the set of 0D scalars specified in Table 1. Figure 5 provides an example comparison of predictions of the $T_e$ and $n_e$ profiles against Thomson measurements for a full shot in the validation dataset, showing accurate prediction across all phases of the shot. This result corroborates previous findings at NSTX-U that neural networks can accurately predict kinetic profiles given a set of similar 0D scalars[42]. Given that this result suggests most of the relevant profile information is implicitly captured by 0D scalars, the profile predictor was disabled prior to running experiments to accelerate training, hence reported predictions of kinetic profiles are not predict-first. This result also suggests a structured data-driven approach to modeling tokamak transport merits further research, in parallel with several ongoing principles-based efforts[43–45]. Another noteworthy feature of this profile predictor that should be explored in future work is its ability to function as a

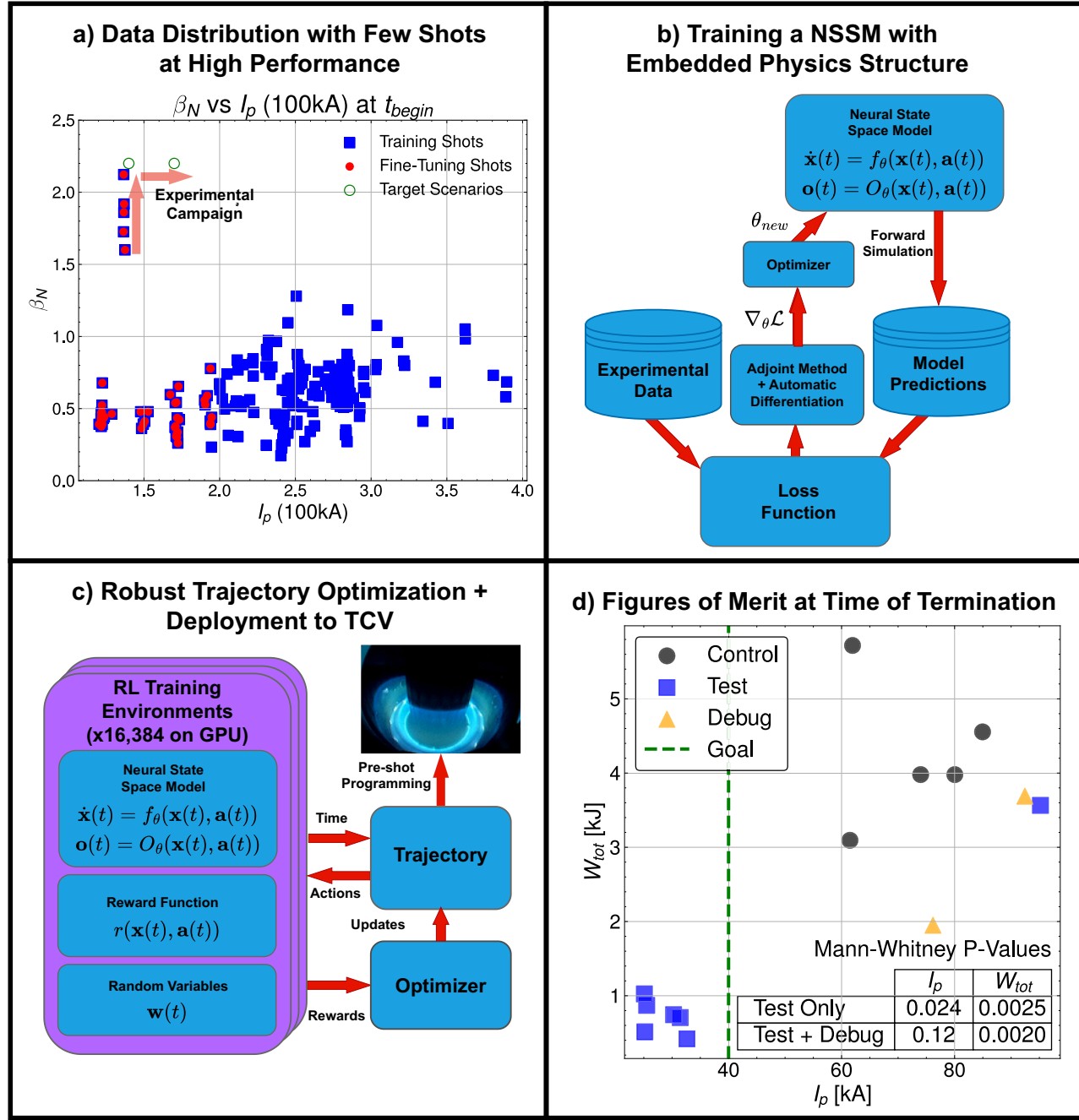

**Fig. 2 | Overview of the methodology and key statistical results. a** The training data distribution, with a modest dataset size at low performance and very few shots in the relevant high-performance regime. The target scenarios for this work at 140 kA and 170 kA with high normalized performance are shown. **b** Depiction of the dynamics model training method, which involves comparing results from forward simulation of an NSSM against experimental data to compute the gradient of loss with respect to model parameters. **c** Depiction of the trajectory optimization process. In addition to the trained dynamics model, the reinforcement learning

(RL) training environment is defined by a reward function specifying the desired goal and a set of random variables that training environments are parallelized against to find a trajectory that has robustness to uncertainties and off-normal events. **d** Scatter plot of plasma current, $I_p$, and stored energy, $W_{tot}$, at time of plasma termination. Bottom-right table shows p-values from the Mann-Whitney U test comparing performance of experimental shots, with and without debug shots included, relative to the control set of all shots in the database with $\beta_N > 1.5$.

Thomson up-sampler, as the input variables are all sampled at a higher time resolution, 1 ms, than the TCV Thomson Scattering system, which takes measurements every 17 ms.

**Preventing VDEs by designing for robustness to control error**
The experiment also clearly highlighted the importance of accounting for control errors when optimizing rampdown trajectories. The rampdowns for the initial shots of the experiment were designed

without accounting for the impact of uncertainty in shape errors on the vertical growth rate $\gamma_{vgr}$. This uncertainty had a highly sensitive effect in TCV #81751, which ended in a VDE. Even though the $\gamma_{vgr}$ at zero control error was tolerable, a small increase in the deviation of the high-field-side (HFS) gap, $g_{HFS}$, from the planned value caused an order-of-magnitude increase in $\gamma_{vgr}$, as shown in Fig. 6.

After #81751, an uncertainty distribution on the gap errors was added to the RL training environments to encourage the optimization of

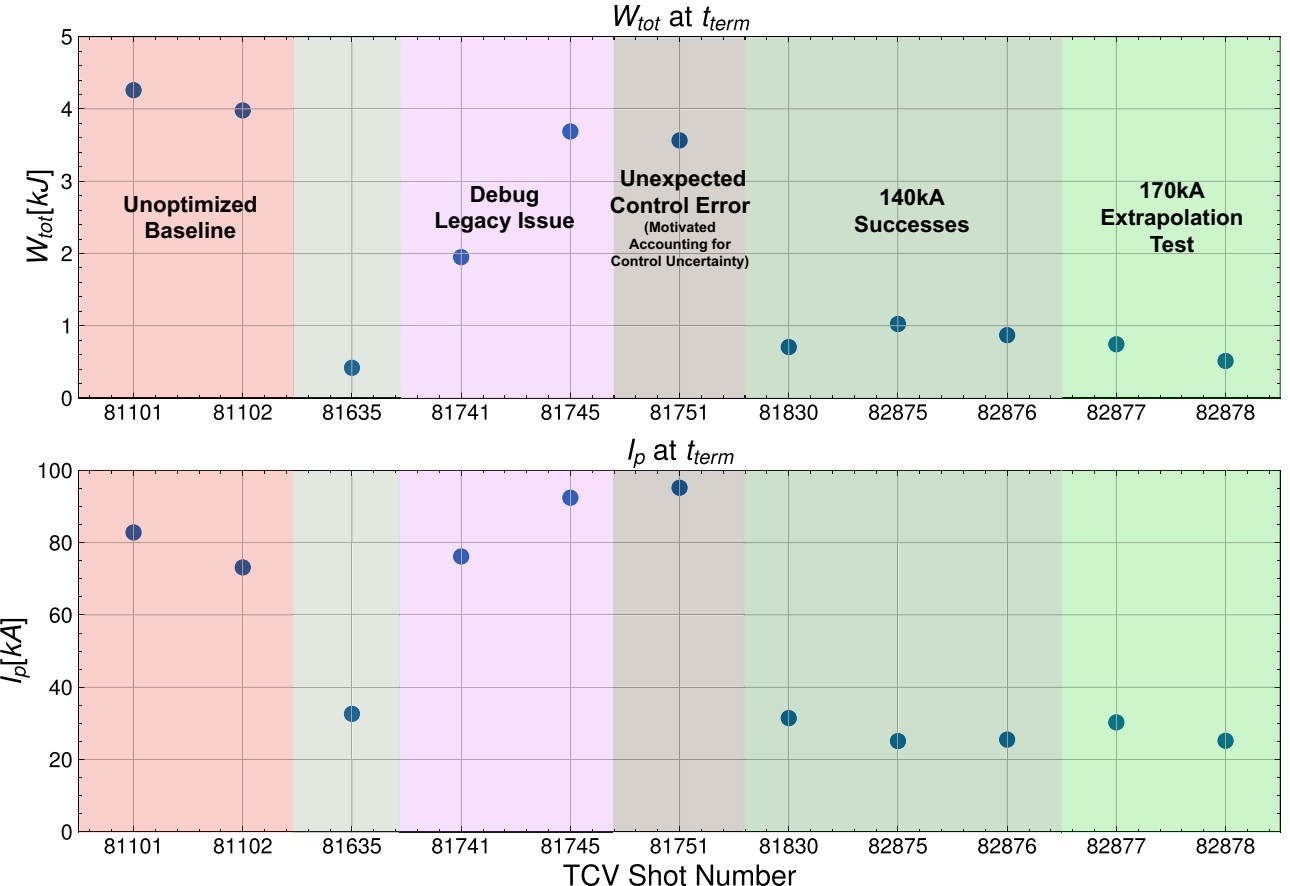

**Fig. 3 | A shot-by-shot breakdown of every shot in the experiment.** The results show the plasma current, $I_p$, and stored energy, $W_{tot}$, at the time of plasma termination, along with additional contextual information. The arbitrary shading is for distinguishing groups of shots described by the labels.

trajectories that succeed despite this control error. The positive impact of optimizing for robustness to this uncertainty was realized with TCV #82875, which experienced similar control errors at similar HFS gap values, but without the large increase in $\gamma_{vgr}$. This increased robustness is likely due to a change in the minor radius trajectory, which decreased the low-field-side (LFS) gap, thus increasing the stabilizing effect of the LFS wall whenever the plasma experiences an unexpected outward shift. Prior work at TCV has shown the importance of managing these gaps for vertical stability[40]. The importance of accounting for this uncertainty is further highlighted by the fact that #82875 is more stable in practice than #81751, despite a higher elongation, the quantity most typically associated with vertical instability. In fact, we can see that #81751, with its lower elongation, does have a lower $\gamma_{vgr}$ than #82875 when the gap error is near zero, but it is also much more sensitive to control errors.

The fact that the trajectory in #82875 is more robust to control error than in #81751 is corroborated by an analysis using the physics-based model for $\gamma_{vgr}$ used in this work[40]. Minor radius variations were introduced to the RL-designed equilibrium trajectories for the two shots, yielding distributions of $\gamma_{vgr}$ trajectories. Figure 6 shows the conclusion that the trajectories in #82875 almost uniformly have lower $\gamma_{vgr}$ than #81751, and stay largely within the soft constraint specified in the RL training process, with the exception of the initial phase of the rampdown process, as the flattop equilibrium has a large $\gamma_{vgr}$.

This result demonstrates that the optimal trajectory for minor radius can differ, with significant consequences, once real-world errors and uncertainties are accounted for. Given that existing studies on rampdown design and optimization for ITER[46] and DEMO[12] find solutions involving large reductions in minor radius, these experimental results motivate the further advancement of techniques that enable trajectory design with robustness to uncertainty.

**Improving terminations by incremental re-training**

Both the NSSM and trajectories were incrementally re-trained on newly generated data from experimental run days, which resulted in more robust and faster rampdowns for the baseline HP scenario, as shown in Fig. 7. The speed of the model enabled re-training of both the model and trajectories in fewer than 10 h total on a single A100 GPU. The unoptimized solution in #81101 involved an NBI power rampdown while keeping constant plasma current, to allow for a decrease in density to avoid the Greenwald limit, a solution which the RL approach initially decided on as well, as shown with #81751, with an even more conservative current ramp and introducing a reduction in $\kappa$. As discussed in, this shot resulted in a VDE, and with the introduction of an uncertainty distribution on $g_{HFS}$, the solution in #81830 resulted in less of a reduction in the minor radius $a_{minor}$, which helped eliminate the $\gamma_{vgr}$ spikes. Subsequent dynamics model training and trajectory optimization resulted in a solution in #82876, which allows for an immediate reduction in $I_p$ without running into a density limit, highlighting the ability for the workflow to assist in gradually making improvements. All optimized trajectories involved a fast initial drop in $P_{NBI}$, followed by a slower ramp phase, although the rates and critical points for the transition differed from shot to shot.

**Predict-first results for the extrapolation test**

Learned dynamics models need not extrapolate far out of distribution to assist with control and trajectory design for net energy tokamaks, as their operations will involve incrementally moving towards higher performance. Thus, they simply need to be able to make reasonable predictions under small extrapolations, and rapidly learn from experiment with as few shots of data as possible. To test the viability of the developed approach in such a setting, we used the learned

**Table 1 | The set of observations predicted by the learned dynamics model in response to action inputs**

| Parameter | Description | Units |
|---|---|---|
| **Actions $a_t$** | | |
| $dI_p/dt$ | Plasma current ramp-rate | MA/s |
| $P_{NBI}$ | Neutral beam injection (NBI) heating | MW |
| $d\kappa/dt$ | Elongation ramp-rate | 1/s |
| $da_{minor}/dt$ | Minor radius ramp-rate | m/s |
| $d\delta/dt$ | Triangularity ramp-rate (zeroed for trajectory optimization) | 1/s |
| $g_{HFS}$ | Gap between the separatrix and the high-field side wall (constant 0.02m and zeroed after NBI off) | m |
| $P_{ECRH}$ | Electron cyclotron resonance heating (ECRH) injected power (zeroed for trajectory optimization) | MW |
| $V_{gas}$ | Primary fueling gas valve voltage (zeroed for trajectory optimization) | Volts |
| **Predicted Observations $o_t$** | | |
| $W_{tot}$ | Plasma total stored thermal energy | kJ |
| $\bar{n}_{e,20}V$ | Line-averaged electron density times volume | $10^{20}$ |
| $f_{GW}$ | Greenwald fraction | - |
| $\beta_p$ | Plasma poloidal beta | - |
| $q_{95}$ | Safety factor at the 95% flux surface | - |
| $\gamma_{vgr}$ | Vertical instability growth rate | 1000 s$^{-1}$ |
| $T_e(\rho)$ | Electron temperature profile (not predict-first) | keV |
| $n_{e,20}(\rho)$ | Electron density profile (not predict-first) | $10^{20}$ m$^{-3}$ |

The time derivatives of certain quantities are chosen as the action to allow the ramp rate to be constrained during trajectory optimization. Note that certain actions are not optimized in this work, but are important to include for the purpose of training the dynamics model; these actions are simply set to zero during the trajectory optimization process. Also note that the set of inputs used for the profile predictor component is $[W_{tot}, \bar{n}_{e,20}, P_{NBI}, P_{ECRH}, I_p, \kappa, a_{minor}, \delta]$.

dynamics model to design trajectories for the extrapolation test scenario, for which zero shots of rampdown data exist in our training dataset for TCV, and generated a priori predictions of the distribution of plasma dynamics during rampdown.

As shown in Fig. 8, experimental results from #82878 largely fell within this distribution, with accurate predictions of the stored energy and density dynamics. Arguably, the largest sources of error came from unreliable control of the plasma shape, contributing to errors in quantities like the rotational transform $\iota_{95} \equiv \frac{1}{q_{95}}$, and also leading to an earlier than expected H-L back-transition. #82878 also started near the density limit, a challenging situation which motivated the introduction of a delay to the $I_p$ ramp in the baseline scenario, but the RL algorithm was able to determine a trajectory to immediately decrease $I_p$, which is desirable, while keeping $f_{GW}$ roughly constant. #82877 fell further out of distribution due to a loss of NBI power, and the presence of a neo-classical tearing mode (NTM) at the beginning of rampdown that did not exist in TCV #82878, as shown by Fig. S5 in the Supplementary Information. Fortunately, these un-modeled ONEs did not take the plasma far enough out of distribution to cause a disruption. As discussed earlier, the profile predictor was removed to help accelerate trajectory optimization, but post-shot evaluation of the profile predictor on the 0D scalars generated by the training environment, shown in Fig. 8, shows reasonable agreement against experimental Thomson measurements.

The results from this experiment demonstrate the ability of the learned dynamics model to make small extrapolations to sufficient accuracy to enable the design of robust disruption-free trajectories via RL, and even the prediction misses in TCV #82877 further emphasize the importance of further advancing the developed methodology to design with robustness to as many ONEs as possible.

## Discussion

Our results demonstrate that the developed approach to learning plasma dynamics can predict the highly transient rampdown phase with a modest dataset and even make small extrapolations to higher performance regimes. The relative sample efficiency of the approach, only requiring five shots in the relevant HP regime, indicates this may be a viable approach for upcoming tokamaks like SPARC and ITER, which will initially operate at low performance before incrementally increasing performance. Developing robust terminations during such incremental campaigns is crucial, as highlighted by the 2020 JET HP campaign, where a 15% increase in plasma current, from 3 and 3.5 MA, raised disruptivity from $\approx$20% to $\approx$50%[11]. Prediction metrics on the validation dataset, as shown in Fig. 4, show that this approach yields accurate predictions for the majority of rampdowns, but the 5% worst cases can involve large prediction errors, meriting further investigation.

The developed RL approach for designing robust trajectories yielded promising improvements in the plasma current and stored energy at the time of termination, with incremental re-training improving the ramp speed. This result represents one of the first successful demonstrations of trajectory design with robustness to real-world uncertainties for tokamaks, which has historically been infeasible due to the computational cost of simulation. A degree of statistical significance is shown, but the sample size is still relatively small; a larger-scale study would more thoroughly determine the efficacy of the approach. Although a large set of uncertainties was accounted for, detailed further in Table S1 in the Supplementary Information, experimental results involved additional uncertainties, such as the NBI failure in #82877, that still need to be addressed to further improve the robustness of trajectories. Robustness to hardware failure is of particular interest for future work, as an exhaustive survey of disruption causes at JET has revealed hardware failure as a significant contributing factor to disruptions[47]. It is also noteworthy that the RL-designed action trajectories tended to be relatively simple, suggesting that the key important ingredient is the fast and parallelized simulation model, as a human expert may be able to find similar trajectories if given access to the simulation model.

To improve the relevance of the developed approach to devices like SPARC and ITER, future work should model additional physics like impurity accumulation and NTM dynamics, both of which are difficult to simulate, partially stochastic, and have been found to be significant contributing factors to disruptions at JET[47]. Accounting for such effects that can drastically change the plasma dynamics may motivate the employment of real-time adjustments to the rampdown trajectory, or the deployment of a library of trajectories as was done in previous simulation studies[48]. Applying the developed approach to learning JET rampdown dynamics would also further inform the application of this approach to SPARC and ITER.

The developed approach also holds promise for full-shot simulation, which ongoing work is investigating[49]. The ability for a neural network to predict kinetic profiles using 0D scalars, demonstrated both in this work and in prior work[42], suggests a data-driven approach may be sufficient for certain control tasks without principles-based transport simulation, which can be extremely computationally expensive and require strong assumptions on edge temperature and density. The ability to deploy accurate, fast, and massively parallel simulators of tokamak plasmas would likely unlock new capabilities for tokamak trajectory and control design, allowing for more reliable access to higher performance plasmas, and ameliorating the risk posed by plasma disruptions to future tokamaks.

## Methods
### The neural state-space model
Learning dynamical systems from data has been a core discipline within control design for decades, including aircraft flight control[20] and simulation[21], and has historically been known as system identification[22,50]. However, due to computational limitations of the

## Model Performance on Initial Validation Set of 131 Shots

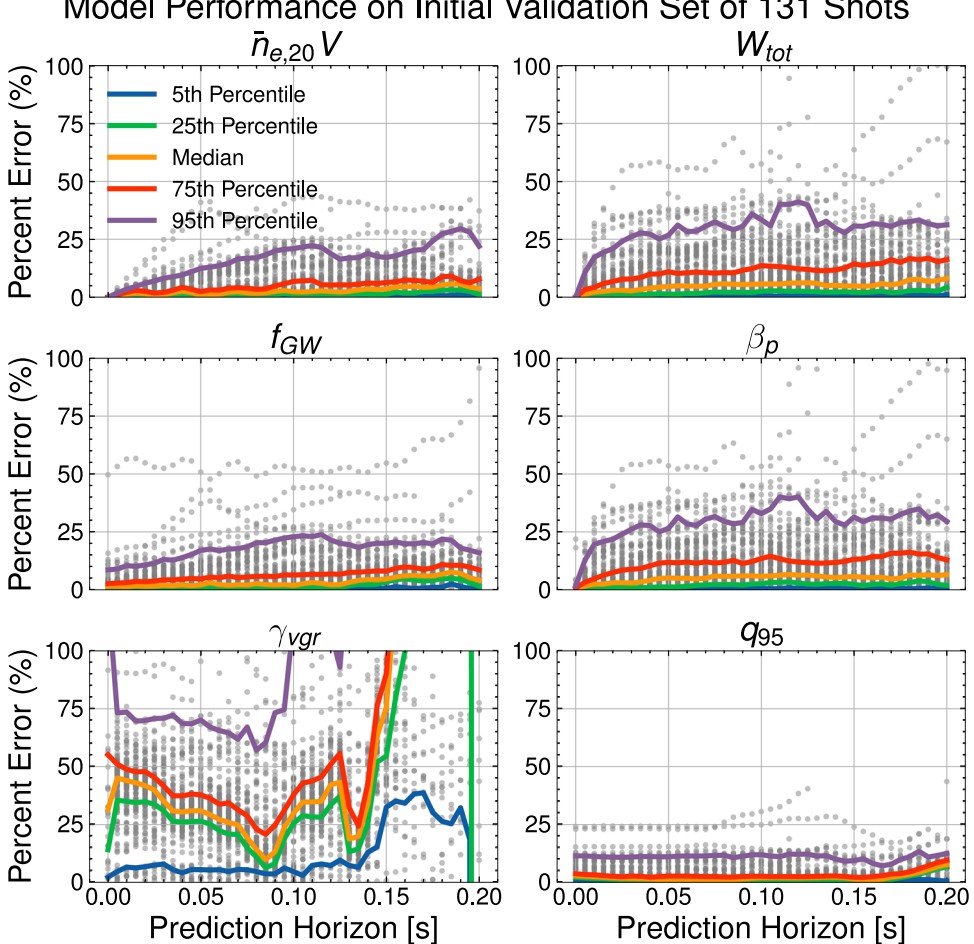

**Fig. 4 | Model validation metrics.** Model prediction accuracy as a function of prediction horizon during rampdowns on the validation set of 131 shots. Both individual data points and percentiles are shown. Shown quantities are the line-averaged electron density times volume, $\bar{n}_{e,20}V$, the total stored energy, $W_{tot}$, the

Greenwald Fraction, $f_{GW}$, the poloidal beta, $\beta_p$, the vertical instability growth rate, $\gamma_{vgr}$, and the safety factor at the 95th flux surface, $q_{95}$. Figure S4 in the Supplementary Information shows similar model performance on the smaller-scale fine-tuning dataset.

time, classical approaches have typically been restricted to linear models, often in the form of linear SSMs:

$$\dot{\mathbf{x}} = A\mathbf{x} + B\mathbf{a} \qquad (1)$$

$$\mathbf{o} = C\mathbf{x} + D\mathbf{a} \qquad (2)$$

Where $A$, $B$, $C$, and $D$ are the matrices to be learned from datasets of observables, $\mathbf{o}$, actions, $\mathbf{a}$, and, possibly, states, $\mathbf{x}$. We note that the controls literature typically uses the notation $\mathbf{y}$ in lieu of $\mathbf{o}$ and $\mathbf{u}$ in lieu of $\mathbf{a}$, reflecting a difference in notation between the controls and RL communities, but here we use RL notation for consistency. In the modern deep-learning learning era, this idea of learning dynamical systems from data was rediscovered from a different perspective, with the advent of the neural differential equation (NDE)[18]:

$$\dot{\mathbf{x}} = NN_\theta(\mathbf{x}) \qquad (3)$$

where it was discovered that, given datasets of $\mathbf{x}$, a neural network, $NN_\theta$, can be used as a system of differential equations that is integrated forward in time with a differential equations solver, and then adjoint back-propagation methods can be used in conjunction with automatic differentiation to determine the gradient of loss with respect to the network parameters $\theta$[16–18]. The introduction of flexible machine

learning frameworks has enabled the development of the field of SciML, based around the core idea of extending NDEs to include physics, and other domain-specific, structure[16,18]. One extension that completes the circle with the classical linear SSM is the NSSM, which reintroduces the concepts of actions and observations:

$$\dot{\mathbf{x}}(t) = f_\theta(\mathbf{x}, \mathbf{a}) \qquad (4)$$

$$\mathbf{o}(t) = O_\theta(\mathbf{x}, \mathbf{a}) \qquad (5)$$

Thanks to the power of new, highly flexible machine learning frameworks such as JAX and the Julia SciML ecosystem, $f_\theta$ and $O_\theta$ can be programmed to include arbitrary combinations of neural networks, physics formulas, and even classical data-driven models such as power laws, a capability which we exploit in this work. The training process of an NSSM is shown in Fig. 9 and involves the simulation of the NSSM forward in time using an initial state $\mathbf{x}_0$ and a time series of actions $\mathbf{a}_{0:T}$ from an experimental database. The error of the simulation results against the experimental data is computed, and adjoint methods and automatic differentiation are used to determine the gradient to reduce the loss. In this work, the differential equation solver package diffrax[17] is used, which includes the integration of multiple adjoint methods with the JAX automatic differentiation system, which allows back-propagation through all differential equation solvers in the package.

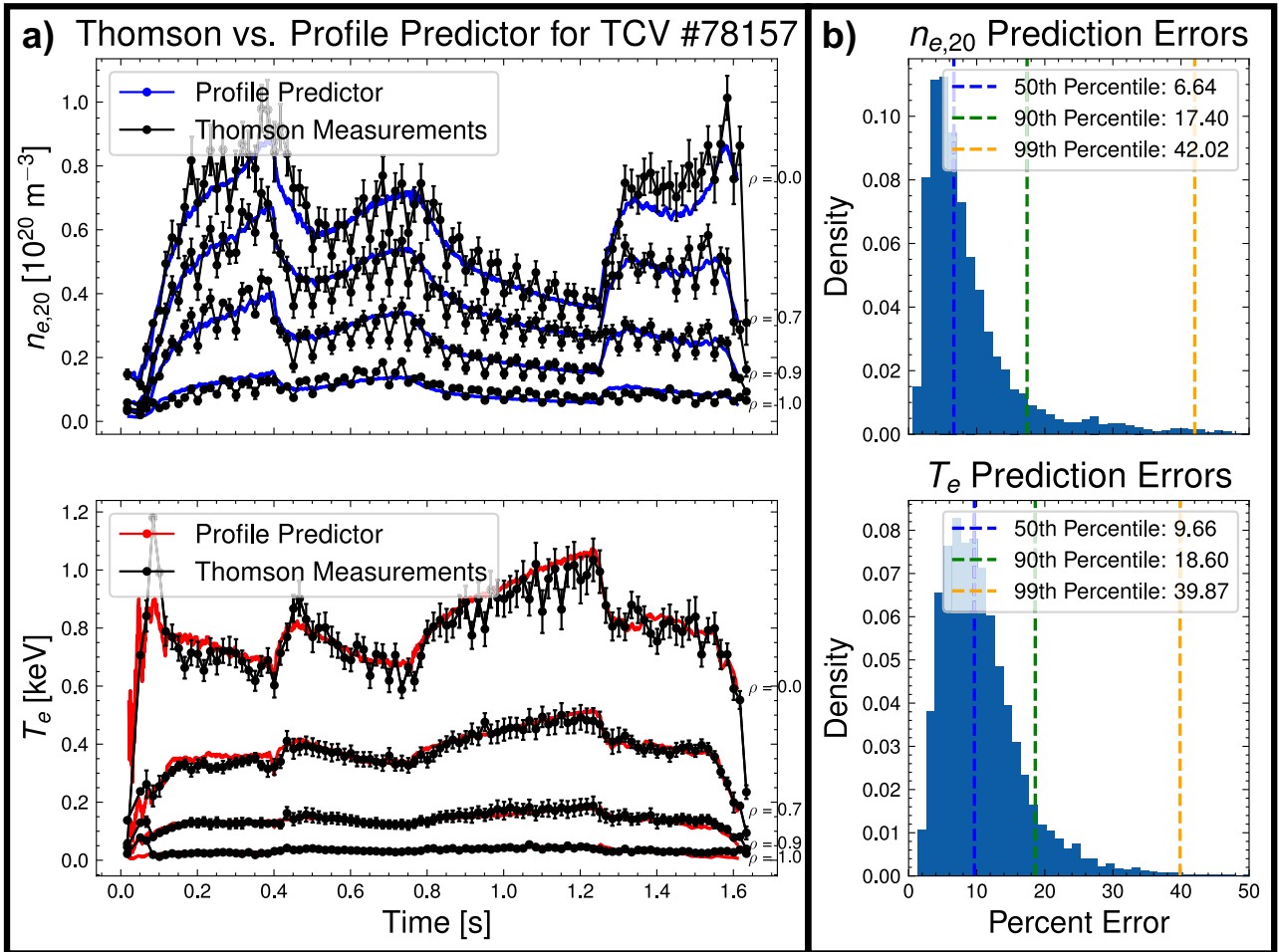

**Fig. 5 | Demonstration of profile predictor performance. a** An example of predictions made by the profile predictor on a validation dataset shot against Thomson measurements, showing both the ability of the model to up-sample Thomson measurements and provide a smoothing effect. Error bars represent two standard deviations. **b** The distribution of prediction percent errors on the validation set for electron density, $n_{e,20}$, and electron temperature, $T_e$. The percent error is defined as the integrated error between the prediction and measurement, normalized to the average value of the profile. For the $T_e$ profile this is: $100 \frac{\int_0^1 |T_{e,Thom}(\rho) - T_{e,pred}(\rho)| d\rho}{\int_0^1 T_{e,Thom}(\rho) d\rho}$. Note that this metric is a pessimistic performance metric as it also captures error due to random measurement noise.

**The dynamics function $f_\theta(\mathbf{x}, \mathbf{a})$.** We begin by defining the following confinement laws:

$$\tau_{n,pred}(\mathbf{x}, \mathbf{a}) = \underbrace{c_n I_p^{c_{I,n}} \bar{n}_{e,20}^{c_{n,n}} P_{input}^{c_{P,n}} \kappa^{c_{\kappa,n}} \epsilon^{c_{\epsilon,n}} |\dot{I}_p|^{c_{I_p,n}} NN_{conf,0}(\mathbf{x}, \mathbf{a})}{} \\ \underbrace{(c_{n,h} \bar{n}_{e,20}^{c_{n,n,h}} P_{input}^{c_{P,n,h}})^{hmode(\mathbf{x}, \mathbf{a})}}{} \quad (6)$$

$$\tau_{E,pred}(\mathbf{x}, \mathbf{a}) = \underbrace{c_E I_p^{c_{I,E}} \bar{n}_{e,20}^{c_{n,E}} P_{input}^{c_{P,E}} \kappa^{c_{\kappa,E}} \epsilon^{c_{\epsilon,E}} |\dot{I}_p|^{c_{I_p,E}}}_{\text{L-mode}} \underbrace{NN_{conf,1}(\mathbf{x}, \mathbf{a})}_{\text{NNcorrection}} \\ \underbrace{(c_{E,h} \bar{n}_{e,20}^{c_{n,E,h}} P_{input}^{c_{P,E,h}})^{hmode(\mathbf{x}, \mathbf{a})}}_{\text{H-modecorrection}} \quad (7)$$

$$hmode(\mathbf{x}, \mathbf{a}) = tanhHeaviside(P_{input} - c_h \bar{n}_{e,20}^{c_{h,n}} a_{minor}^{c_{h,a}}) \quad (8)$$

$$tanhClip(x) \equiv tanh\left(\frac{2k}{max - min}(x - center)\right)\frac{max - min}{2} + center \quad (9)$$

$$tanhHeaviside(x) \equiv \frac{1}{2}(tanh(kx+1)) \quad (10)$$

where the parameters to be learned include all coefficients $c_*$ and neural network parameters. The laws are structured to multiply a portion corresponding to L-mode, a neural network correction factor, and an H-mode correction factor. The L-mode term reflects standard confinement scalings, but with the introduction of a $\dot{I}_p$, which was found to help better capture the short-term effects of ramping plasma current. The neural network output includes a tanhClip final activation that constrains its output to the range [0.75, 1.25], thus controlling the maximum adjustment the network is allowed to make. The hmode function includes a tanhHeaviside function, which provides a smooth transition between one to zero once $P_{input}$ falls below the learned back-transition threshold, which is structured to reflect the Martin scaling[51]. Note that the use of the hmode function output as a power deactivates the H-mode correction term once hmode transitions from one to zero. While, in principle, the neural network should be able to learn the effects of H-mode implicitly, we found that adding an explicit H-mode correction factor helped improve model predictions in our low-data regime. The $k$ factor controls the smoothness of both the tanhClip and tanhHeaviside functions.

These confinement laws are integrated as a part of the following 0D energy and particle balance equations, which is a model that blends

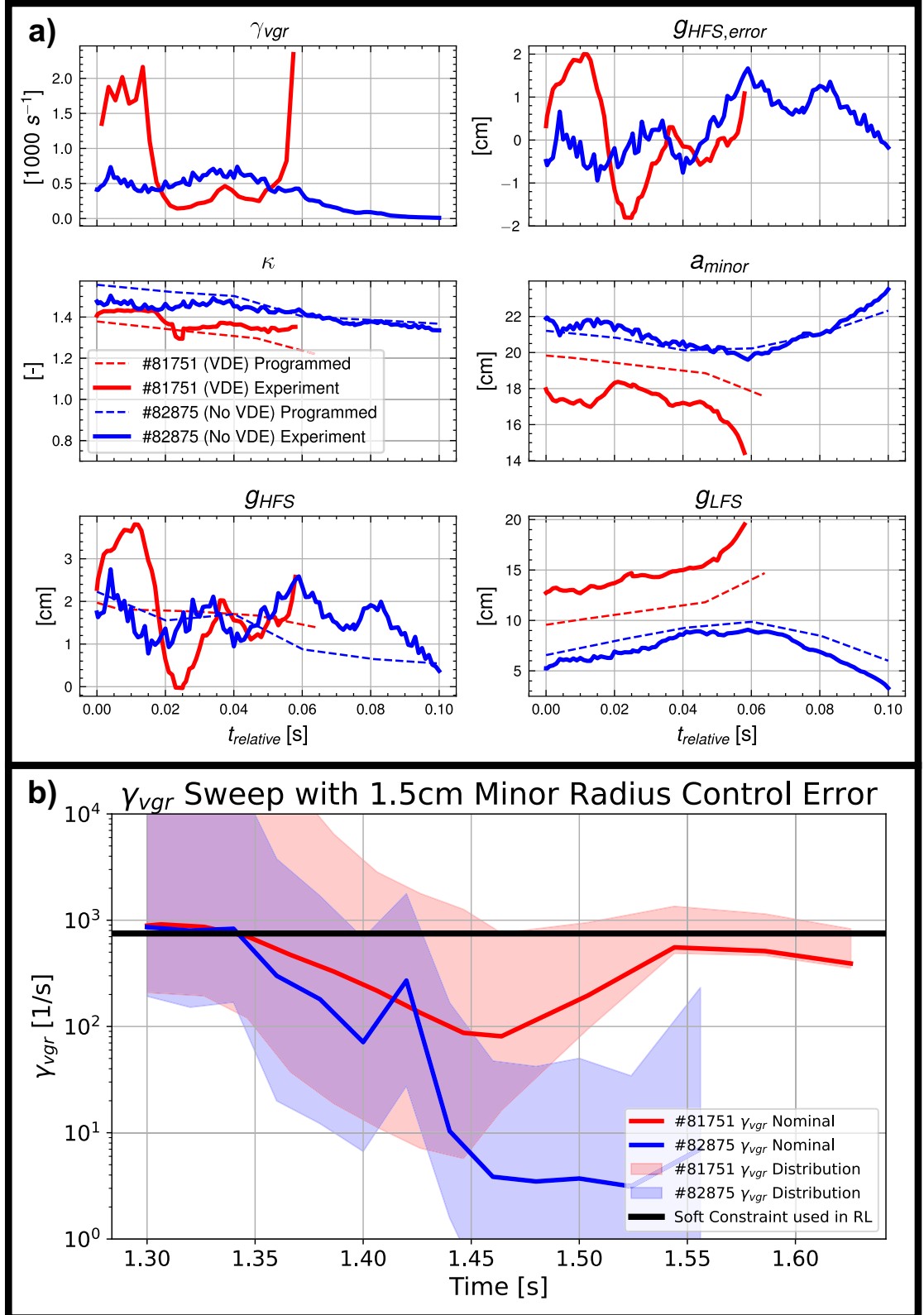

**Fig. 6 | Demonstration of increased robustness to control uncertainty. a** A comparison of a shot designed prior to adding control uncertainty, #81751, and after, #82875. We see that the vertical instability growth rate, $\gamma_{vgr}$, is highly sensitive to control error in the high-field side gap, $g_{HFS,error}$, in #81751, but similar control errors experienced in #82875 result in negligible changes in stability. Time shown, $t_{relative}$, is the time relative to 1.4 s for #81751 and 1.3 s for #82875 to align the moment the two shots experience similar control errors. Additional quantities shown are the plasma elongation $\kappa$, minor radius $a_{minor}$, high-field side gap, $g_{HFS}$, and low-field side gap, $g_{LFS}$. **b** $\gamma_{vgr}$ distributions under 1.5 cm of minor radius control error for #81751 and #82875. Control error was simulated by introducing variations in the control points provided to the free-boundary equilibrium solver, FBT[69,75], used for shot preparation. The resulting equilibria were then input into the $\gamma_{vgr}$ computation method used in this work[40].

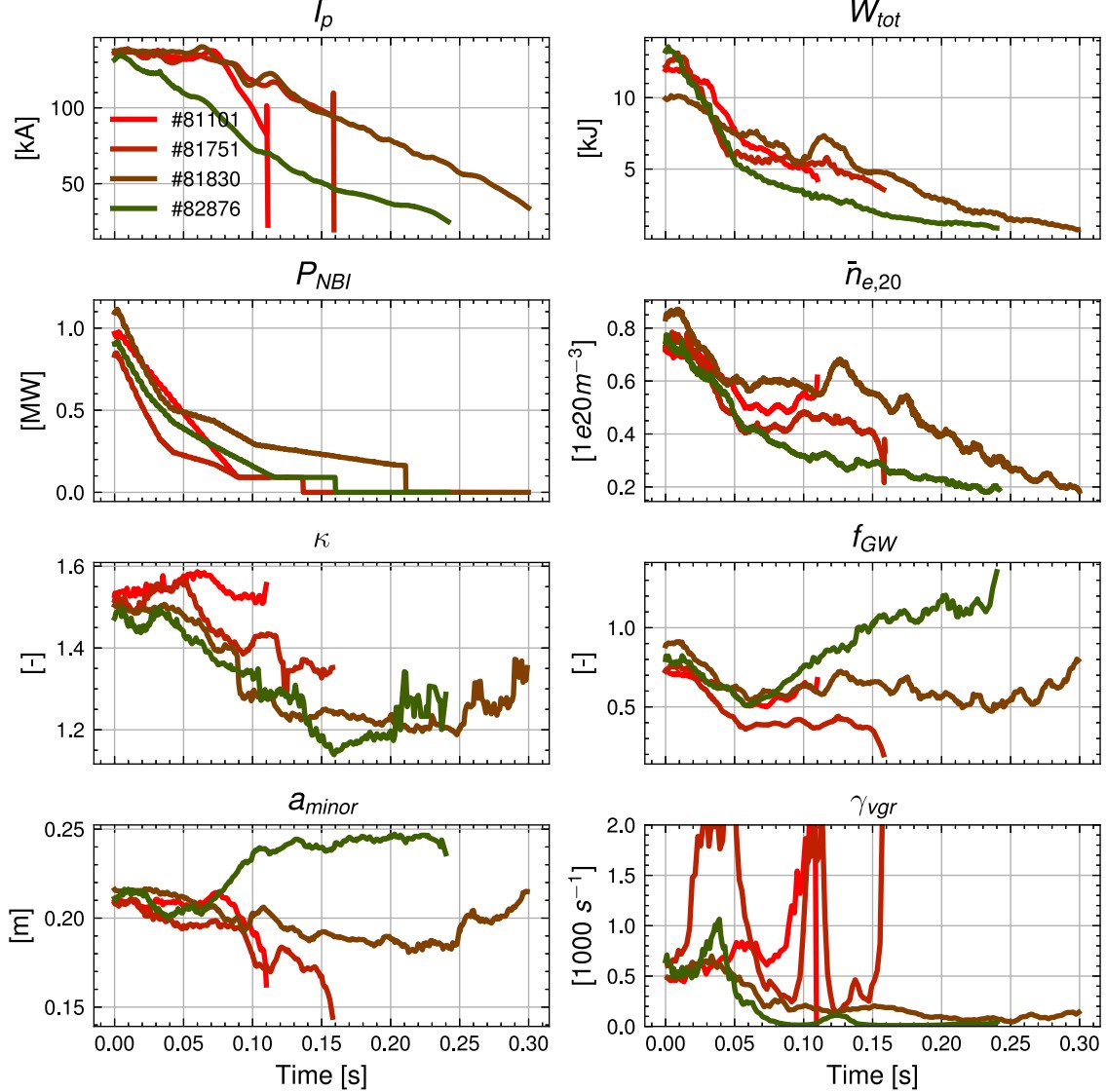

**Fig. 7 | Incremental improvements for the baseline high-performance scenario.** Experimental traces of key actuators are shown (left), including plasma current, $I_p$, neutral beam power inject, $P_{NBI}$, elongation, $\kappa$, and minor radius, $a_{minor}$. Additional relevant quantities (right) include the stored energy, $W_{tot}$, line-averaged electron density, $\bar{n}_{e,20}$, Greenwald fraction, $f_{GW}$, and vertical instability growth rate, $\gamma_{vgr}$. Time is set relative to the beginning of the termination phase. The plasma current, $I_p$, and stored energy, $W_{tot}$, trajectories show improvement in rampdown speed and disruptivity as the experiment progressed.

simple physics principles, power laws, and neural networks:

$$\frac{dW_{tot}}{dt} = -\underbrace{\frac{W_{tot}}{\tau_{E,pred}}}_{\text{Transport}} + \underbrace{I_p^2 NN_{ohm,rad,0}(\mathbf{x}, \mathbf{a})}_{\text{OhmicHeating}} - \underbrace{\bar{n}_{e,20} V NN_{ohm,rad,1}(\mathbf{x}, \mathbf{a})}_{\text{RadiatedPower}}$$
$$+ \underbrace{P_{NBI} + P_{ECRH}}_{\text{Aux.Heating}} \tag{11}$$

$$\frac{d(\bar{n}_{e,20} V)}{dt} = -\underbrace{\frac{\bar{n}_{e,20}}{\tau_{n,pred}}}_{\text{Transport}} + \underbrace{c_{NBI} P_{NBI}}_{\text{NBIFueling}} + \underbrace{c_{gas,0} \sigma(c_{gas,1} V_{gas} + c_{gas,2})}_{\text{GasValveFueling}}$$
$$+ \underbrace{NN_{wall}(\mathbf{x}, \mathbf{a}) \exp^{-c_{wall} g_{HFS}}}_{\text{WallEffects}} \tag{12}$$

where $\sigma$ is the sigmoid function, $NN_{ohm,rad}$ is a network that predicts two quantities; the first is multiplied by $I_p^2$ to serve as an Ohmic heating term, and the second is multiplied by density and volume to serve as the radiated power term. $NN_{wall}$ is included to account for possible wall

fueling effects when in a limited configuration, and is multiplied by an exponential in the HFS gap to deactivate it when diverted. Additional simple constants are included to account for fueling from both NBI and gas puffing. We note that, in both cases, the included terms do not capture important state dependencies and time delays, but they proved sufficient for this use case. The dynamics of density times volume are predicted; in cases where density itself is used (e.g., to compute the Greenwald Fraction), the following volume approximation is used to recover density:

$$V \approx 2\pi R^2 \epsilon^2 \kappa \left( \pi - \left( \pi - \frac{8}{3} \right) \epsilon \right) \tag{13}$$

Since time derivatives of quantities, $\dot{I}_p, \dot{\kappa}, \dot{a}_{minor}, \dot{\delta}$ are used as actions, their integrated values are also added as state variables with trivial dynamics.

**The observation function $O_\theta(\mathbf{x}, \mathbf{a})$.** The observation function consists of several components: a NN predictor for $\gamma_{vgr}$, a profile predictor, and

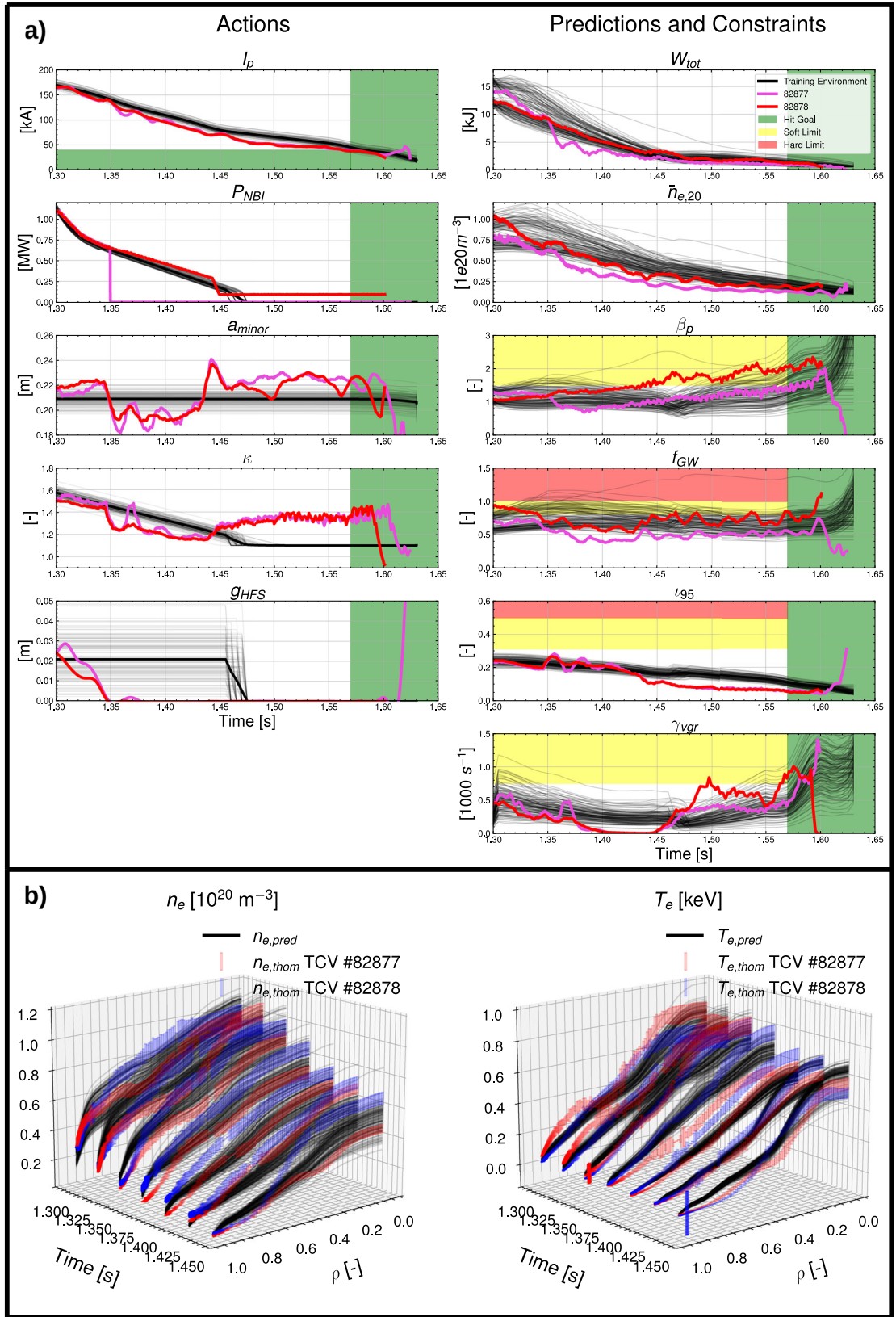

**Fig. 8 | A priori predictions and experimental results for the extrapolation test scenario. a** Action trajectories and a priori predictions of plasma dynamics during rampdown, compared to experimental results from TCV #82877 and #82878. The RL training environment accounts for uncertainty in actuation with distributions on the action trajectory; the average of the distribution (in solid black) is used for shot programming. Control of the plasma shape proved to be a challenge for this phase, an issue also observed in previous rampdown studies[6,8]. Shown action variables include the plasma current, $I_p$, neutral beam injected power, $P_{NBI}$, minor radius, $a_{minor}$, elongation, $\kappa$, and high-field side gap, $g_{HFS}$. Shown predictions and constraints include the stored energy, $W_{tot}$, the line-averaged electron density, $\bar{n}_{e,20}$, poloidal beta, $\beta_p$, Greenwald fraction, $f_{GW}$, rotational transform at the 95% flux surface, $\iota_{95}$, and vertical instability growth rate, $\gamma_{vgr}$. **b** Post-hoc predictions of electron temperature, $T_e$, and density, $n_e$, profiles compared to Thomson Scattering measurements.

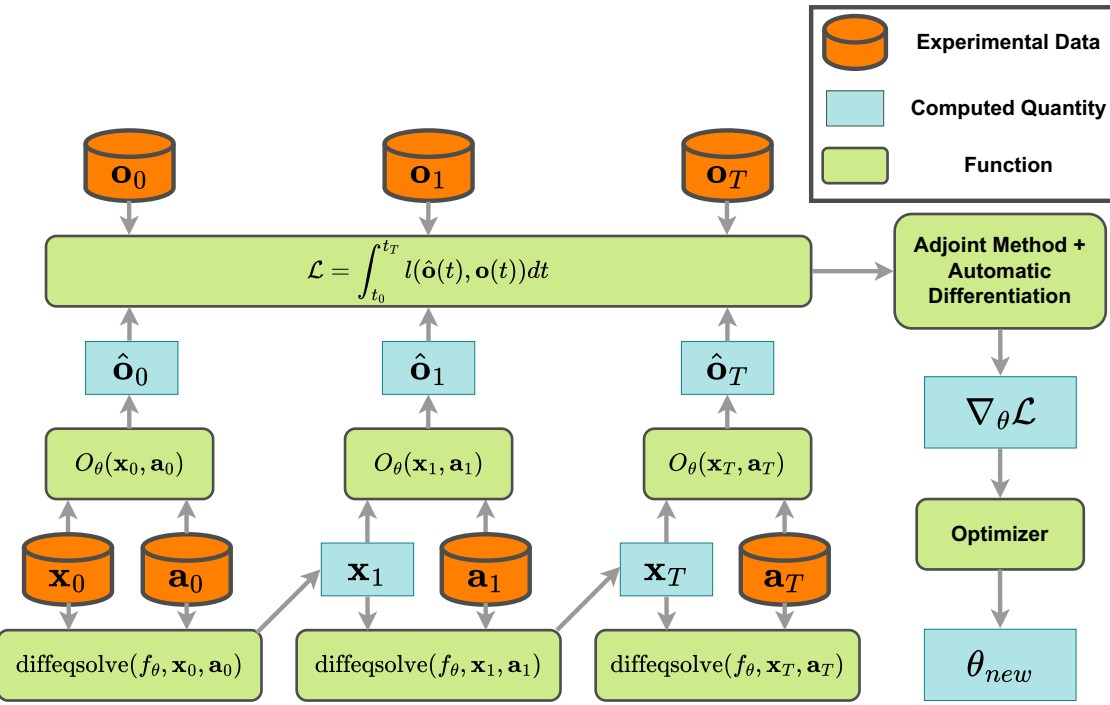

**Fig. 9 | Depiction of the neural state-space model (NSSM) training methodology.** The NSSM, defined by the dynamics function $f_\theta$ and observation function $O_\theta$ with parameters $\theta$, is simulated forward in time, given an initial state $\mathbf{x}_0$ and an action trajectory $\mathbf{a}_{0:T}$, to generate a sequence of simulated observations, $\hat{\mathbf{o}}_{0:T}$. The simulated observations are compared with experimental observations via the loss functional $\mathcal{L}$, which is defined as the time-integrated value of an instantaneous loss function $l$. Adjoint methods in diffrax[17] and JAX automatic differentiation then yield the gradient of model parameters with respect to loss, $\nabla_\theta \mathcal{L}$, which allows the optimizer to update the parameters $\theta$.

simple physics formulae to compute derived quantities:

$$
O_\theta(\mathbf{x}, \mathbf{a}) \begin{cases}
\beta_p = \frac{8}{3} \frac{W_{tot}}{\mu_0 R_0 I_p^2} & \text{(14c)} \\
f_{GW} = \frac{\bar{n}_{e,20} \pi a_{minor}^2}{I_{p,MA}} & \text{(14d)} \\
q_{95} = \frac{4.1 a_{minor}^2 B_0}{R_0 I_{p,MA}} \left(1 + 1.2(\kappa-1) + 0.56(\kappa-1)^2\right)\left(1 + 0.09\delta + 0.16\delta^2\right)\frac{1+0.45\delta\epsilon}{1-0.74\epsilon} & \text{(14e)} \\
\gamma_{vgr} = NN_{vgr}(\mathbf{x}, \mathbf{a}) & \text{(14f)} \\
T_e(\rho), N_e(\rho) = NN_{prof}(\mathbf{x}, \mathbf{a}) & \text{(14g)}
\end{cases}
$$

where $\beta_p$ is computed in accordance with the LIUQE definition[52], $f_{GW}$ is the usual Greenwald Fraction[9], $q_{95}$ is the approximation given in[53] with the squareness factor set to 1, $NN_{vgr}$ is a multi-layer perceptron (MLP) with GELU activation and a scaled sigmoid output, and $NN_{prof}$ is a neural-operator-based profile predictor, discussed further in the next subsection.

**Neural-operator-based profile predictor.** Prior work at NSTX-U trained a neural network to successfully predict kinetic profile shapes using their averages plus zero-dimensional control parameters such as plasma current, shaping, and auxiliary heating. Building upon this prior work, we show that, on TCV data, kinetic profiles can be predicted to reasonable accuracy with a neural network using the stored energy $W_{tot}$, line-averaged electron density $\bar{n}_{e,20}$, and control parameters. The key implication is that accurate prediction of the time-dependent dynamics of just two scalars, $W_{tot}$ and $\bar{n}_{e,20}$, implies reasonable prediction of the dynamics of kinetic profiles.

We leverage methods developed by the neural operator[54,55] literature, which has found success for solving machine learning problems in scientific domains involving PDEs. Letting $f_{in}$ denote an input function and $f_{out}$ denote an output function, a neural operator $\mathcal{F}$ parameterized by $\theta$ maps an input function to an output function:

$$f_{out} = \mathcal{F}_\theta(f_{in}) \tag{15}$$

In practice, the functions involved are approximated using a set of basis functions; thus, the practical implementation results in a neural network operating on basis function coefficients. In this work, we make use of cubic B-spline basis functions to represent the kinetic profiles:

$$T_e(\rho) = \sum_{i=1}^{n_{basis}} c_{T,i} B_{i,3}(\rho) \quad n_{e,20}(\rho) = \sum_{i=1}^{n_{basis}} c_{n,i} B_{i,3}(\rho) \tag{16}$$

And we predict these profiles using a set of 0D scalars, where every scalar is a control parameter except stored energy $W_{tot}$ and $\bar{n}_{e,20}$. The full set of input and output parameters is specified in Table 1. During training, the $\rho$ grid corresponding to the dataset is chosen to evaluate the basis functions, but arbitrary alternative grids can be used during inference time.

**Training methods.** Training of the NSSM involved two stages. First, $NN_{ohm,rad}$, $NN_{vgr}$, and $NN_{prof}$ are trained independently of the rest of the model on time-independent samples to predict their respective quantities. These "pretrained" models are then integrated into the NSSM, where they are further trained jointly with the rest of the model through the time-dependent process specified in Fig. 9. The AdamW optimizer[56] with an exponential decay learning rate schedule is used for every training run. All NNs used in the dynamics function $f$ and $NN_{vgr}$ are simple MLPs with GELU[57] activations on the hidden state and tanhClip functions as final activations to constrain their outputs to reasonable ranges. The profile predictor is further detailed earlier in the methods. Hyperparameters for the optimizer and model sizes are optimized via Bayesian Optimization using the method implemented in the Weights and Biases platform[58], which was used in this work for experiment tracking. The final set of hyperparameters is detailed in Table S2 in the Supplementary Information.

## Training data distribution

The dataset used for training models in this work consists of the 442 most recent shots with rampdowns that are at least partially complete and have sufficient diagnostic availability, gathered with Disruption and Event analysis framework for FUSion Experiments[59]. The initial training phase involved training on 311 shots of data, with the rest of the dataset used for validation. After the initial training phase, the model is further trained on a fine-tuning dataset of 44 shots. During this phase, all of the model weights except those in the $\tau_E$ and $\tau_N$ hybrid confinement laws described in are frozen. As shown in Fig. S8 in the Supplementary Information, the dataset consists of only five shots of data anywhere near the relevant HP region.

## Reward function

The reward function is designed to balance the priority of achieving a low plasma current and energy against the risk of disrupting the plasma, and is given by:

$$r(\mathbf{x}(t), \mathbf{a}(t)) = \underbrace{-c_{time}}_{\text{Penalty for time}} - \underbrace{c_W W_{tot}(t) - c_{I_p} I_p(t)}_{\text{Penalty for current and energy}} - \underbrace{\sum_{i=1}^{n_{soft}} c_{soft} s_i(\mathbf{x}(t))}_{\text{Soft chance-constraints}}$$
$$- \underbrace{\sum_{i=1}^{n_{hard}} c_{hard} h_i(\mathbf{x}(t))}_{\text{Hard chance-constraints}} \qquad (17)$$

The reward function is active for every time step before hitting the goal state or the maximum allowed training episode time. The goal state is chosen to be a stored energy of 500 J and a plasma current of 40 kA, as, for the 170 kA extrapolation test scenario, 40 kA approximately corresponds to the relative plasma current for an ITER 15 MA benign termination, which is defined as 3 MA[60]. A constant penalty term is active for every time step before achieving the goal to encourage time minimization. In addition, penalty terms that scale with plasma current and energy are included to further prioritize moving towards a safer state. To avoid disruptive limits such as high Greenwald fraction during the rampdown, penalty terms are added for states that violate user-specified constraints on key quantities correlated with disruptions.

One challenge with specifying constraint limits is the difference in severity of violating different constraints, and the, at times, weak correlations between physical quantities and disruptions. To address this issue, we partition constraints into "soft" constraints, which incur a small penalty to discourage, but not forbid, the algorithm from finding solutions that violate these limits, and "hard" constraints, which incur a large penalty to strictly enforce constraint violation. We note that while methods in the constrained optimization literature often mathematically express constraints separately from the objective function being optimized, most practical implementations of constrained optimization algorithms enforce constraints by rewriting constraints as penalty terms in the objective function[61,62], an approach we also adopt. Stochastic optimization across a distribution of outcomes introduces a challenge: trying to avoid limits for every scenario will likely result in excessively slow and conservative solutions[63], which itself poses its own risk. To address this challenge, we utilize chance-constraints, a technique often utilized in the autonomous driving literature[64,65], and only activate the constraint if violation probability exceeds a certain threshold. In this set of experiments, this threshold is chosen as 5%. Reward function parameters used for the final four shots are shown in Table S2 in the Supplementary Information.

## Uncertainty model

In experimental reality, the time evolution of plasma dynamics is highly nonlinear and subject to considerable amounts of uncertainty, as evidenced by the two same-scenario shots shown in Fig. 8, which begin at drastically different initial conditions. To design trajectories that have robustness to large variability and ONEs, we defined an uncertainty model for the RL training environments, and sampled from this uncertainty model for each training environment used during training. The uncertainty model includes random variables for both the initial state of the plasma during rampdown and disturbances/model uncertainties that affect the time-varying plasma dynamics. To account for the fact that accidental H-L back-transition implies the initial state of the plasma may start in either H or L-mode, the initial state distribution is modeled as a bi-modal mixture model, with a 50% chance of any given RL training environment starting in either H or L-mode. In some cases, uncertainty distributions could easily be quantified from past experimental data (such as tracking error in the plasma current), or from model prediction accuracy (such as $\gamma_{ugr}$), but in other cases, the distribution was chosen in an ad-hoc fashion, upon identifying additional sources of uncertainty in the experiment. Table S1 in the Supplementary Information summarizes the random variables, parameterized distributions, and quantification methods used in this work. As discussed in the section, this uncertainty model proved to be non-exhaustive in the experiment. In addition, the uncertainty model employed does not account for time-varying fluctuations in uncertain variables; future work should employ time-varying stochastic processes. Both of these limitations further highlight the need to advance experimental uncertainty quantification and robust control in the context of fusion plasma control.

## RL methods

Standard RL problems involve optimizing a policy $\pi$ to map observations to actions:

$$\mathbf{a} = \pi(\mathbf{o}) \qquad (18)$$

from this perspective, trajectory optimization can be viewed as policy optimization where the only observable is time:

$$\mathbf{a} = \pi(t) \qquad (19)$$

Given that time is the only observable, but there exist different physical conditions in the parallel training environments that are unobservable to the policy, the reward maximization process yields a trajectory that is designed to succeed across the different conditions specified in the subsection. After an initial trial with Proximal Policy Optimization[66], we found OpenAI-ES, an evolutionary strategy (ES) designed for policy optimization, to work better in practice[67]. This is possibly explained by the theoretical analysis given in the paper introducing OpenAI-ES, which suggests that RL problems with long time horizons and actions that have long-lasting effects may be better solved with ES approaches than the dominant paradigm of policy gradient methods[67]. The policy $\pi$ was parameterized by an MLP with two hidden layers of width 64 and used ReLU activations with a hyperbolic tangent final activation to constrain the action space. A hyperparameter sweep of the architecture was not employed, and it would be worthwhile to investigate for future work.

## Deployment to TCV

Shape trajectories determined via RL were mapped to last-closed-flux-surface control points via re-scaling of the flattop shape for the diverted phase, and using an analytic formula in the TCV MGAMS[68] algorithm for the limited phase. Feed-forward coil currents and voltages to achieve the desired plasma current and shaping trajectories were then determined with the free-boundary equilibrium code FBT and shot

preparation algorithm MGAMS[68,69], and the $P_{NBI}$ trajectory was programmed into the TCV supervisory control system SAMONE[70,71].

## Data availability

Data to generate figures found in this paper are available in the source code repository at https://doi.org/10.5281/zenodo.16621120[72]. The complete dataset used for training and validation can be obtained by contacting A.P. or A.M.W.

## Code availability

Source code is provided at https://doi.org/10.5281/zenodo.16621120[72].

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

## Acknowledgements

A.M.W. and C.R. were funded in part by Commonwealth Fusion Systems. The work of A.P., O.S., A.V., C.G., A.M., Y.P., C.V., F.F., and S.M. has been carried out within the frame-work of the EUROfusion Consortium, via the Euratom Research and Training Programme (Grant Agreement No 101052200 - EUROfusion) and funded by the Swiss State Secretariat for Education, Research, and Innovation (SERI). Views and opinions expressed are however those of the author(s) only and do not necessarily reflect those of the European Union, the European Commission, or SERI. Neither the European Union nor the European Commission nor SERI can be held responsible for them. The authors would like to acknowledge the Engaging cluster, managed by the MIT Office of Research Computing and Data, which was used in this work for model training.

## Author contributions

Allen M. Wang led the project, developed the dynamics modeling approach and RL problem formulation, and led the writing of the paper. Alessandro Pau developed the machine learning dataset, high-performance integrated control scenario, and integration of the approach into TCV. Oswin So and Charles Dawson worked with Allen M. Wang to develop the RL training environments and methods. Olivier Sauter enabled TCV integration and identified and debugged the radial observer issue. Cristina Rea managed the collaboration, advised the project on disruptions and machine learning, and contributed to manuscript drafting and revisions. Mark D. Boyer advised the project from a controls perspective and motivated the approach taken for dynamics modeling. Anna Vu and Cristian Galperti enabled integration with the integrated control system. Chuchu Fan advised Allen M. Wang, Oswin So, and Charles Dawson on controls and RL approaches. Antoine Merle developed and assisted with MHD equilibrium codes. Yoeri Poels and Cristina Venturini helped with the TCV dataset. Federico Felici developed the equilibrium and plasma control infrastructure that enabled this work and advised the development of the growth rate calculation method. Stefano Marchioni developed the growth rate calculation method used in this work. The TCV team enabled the experiments done in this work by developing, maintaining, and operating the TCV tokamak.

## Competing interests

The authors declare the following competing interests: M.D.B. is an employee of Commonwealth Fusion Systems, and A.M.W. and C.R. are

funded in part by Commonwealth Fusion Systems. The remaining authors have no competing interests to declare.
