## [Transparent Peer Review file · Nature Communications]

Learning Plasma Dynamics and Robust Rampdown Trajectories with Predict-First Experiments at TCV

Corresponding Author: Mr Allen Wang

Version 0:

Reviewer comments:

Reviewer #1

(Remarks to the Author)

The work revolves around the use of reinforcement learning (RL) for the optimization of the ramp-down trajectory at TCV, an experimental device in Switzerland. The idea of RL is that the model can learn how to interact with the system by trial and error and requires a sufficiently accurate model of the system. In this case, also the system is approximated by a Neural State Space Model (NSSM), that is a Neural Network which can approximate the state space representation mixing a data driven loss and numerical integration of the (net+system) dynamics.

The work is interesting but due to the scarcity of experiments, the successfully designed rampdowns are only 5, out of a total of 9 optimized ones. For this reason, a further analysis and replication of the experiments is recommended for making robust conclusions.

Moreover, I have the following comments on the work:

NSSM and training environment for the RL system

The article is quite dense, and the authors explain that one model (the NSSM) serves as the training environment for the RL model, however another neural model to predict the temperature and density profiles is mentioned as part of the NSSM.

From section 2.2:

“The NSSM was initially developed with a neural network predictor for the kinetic profiles on the full ρ grid, and initial training runs found that the profile predictor can accurately predict kinetic profiles given the set of 0D scalars listed in Table 2.

Figure 5 provides an example comparison of predictions of the T_e and n_e profiles against Thomson measurements for a full shot in the validation dataset, showing accurate prediction across all phases of the shot. This result corroborates previous findings at NSTX-U that neural networks can accurately predict kinetic profiles given a set of similar 0D scalars³⁴. Given this result suggests most of the relevant profile information is implicitly captured by 0D scalars, the profile predictor was disabled prior to running experiments to accelerate training, hence reported predictions of kinetic profiles are not predict-first. A noteworthy feature of this profile predictor that should be explored in future work is its ability to function as a Thomson up-sampler, as the input variables are all sampled at a higher time resolution, 1ms, than the TCV Thomson Scattering system, which takes measurements every 17ms.”

If the predictor has not been adopted, I suggest removing the parts connected with the profile predictor (e.g section 4.1.3, Figures 5, 9) to avoid confusion, otherwise I would suggest indicating for how many discharges the predictor has been used to train the RL method and if there was any benefit from this approach.

How were the discharges to train the NSSM model selected? What are the variations in terms of the (observation, action) space?

Connected to the previous point, it seems that the vertical instability growth rate (γ_{vgr}) reconstruction by the NSSM is very poor, even with a very low prediction Horizon (Figure 4). Since this parameter is connected to vertical plasma stability, this would explain the VDE event in shot #81751. How large is the uncertainty assumed for the γ_{vgr} ?

An interesting point could be made by running the control discharges in the NSSM and showing that they were close to disruptive boundaries/expected to disrupt. This would show, despite the low number of experiments, that the NSSM can well represent the system.

Is it possible to interpret the strategies adopted by the RL model by showing which parameters are kept far from critical boundaries?

Other comments

Some other works related to the use of RL in fusion are missing, as applications for ITER [1] and EAST [2]. Please add some details on the model adopted (number of layers, neurons) and on how they have been selected/optimized. I tried running the code and installing all the dependencies using poetry, but I could not manage to make it work. Most of the python notebooks seem to reproduce the figure and the main results, I did not find the scripts for training the models.

[1] S. Dubbioso, G. De Tommasi, A. Mele, G. Tartaglione, M. Ariola, and A. Pironti, 'A Deep Reinforcement Learning approach for Vertical Stabilization of tokamak plasmas', Fusion Engineering and Design, vol. 194, p. 113725, Sep. 2023, doi: 10.1016/j.fusengdes.2023.113725.

[2] G. De Tommasi, S. Dubbioso, Y. Huang, Z. P. Luo, A. Mele, and B. J. Xiao, 'A RL-based Vertical Stabilization System for the EAST tokamak', in 2022 American Control Conference (ACC), Jun. 2022, pp. 5328–5333. doi: 10.23919/ACC53348.2022.9867499.

(Remarks on code availability)

Partial review of the code due to the difficulties in installing all the proper libraries

The readme file instructs on using poetry, but after finding the correct version of poetry I ran into an error:

Directory TokaGym\submodules\contrax for contrax does not seem to be a Python package

Reviewer #2

(Remarks to the Author)

General Comments:

This study leverages deep learning (NSSM) to predict plasma ramp-down dynamics in the TCV tokamak and applies reinforcement learning (RL) to design an optimal trajectory for stable termination during this phase. Notably, ramp-down in fusion reactors is physically and technically prone to instability, often associated with plasma disruptions. Thus, ensuring stable control and termination in this phase is critical for future large-scale devices such as ITER. This study combines a predictive model and RL on TCV to design a stable plasma termination trajectory and demonstrates robust outcomes through incremental re-training with actual experiments. This represents a significant milestone in fusion research. I believe that, provided the following comments are adequately addressed, the manuscript could be suitable for publication in Nature Communications.

Major Comments:

1. While the significance and performance of the study are commendable, the storyline in the figures and descriptions is not very clear. Since Nature Communications is a multidisciplinary journal, the content should be understandable even to readers who are not fusion experts. From a non-expert's perspective, when reading the early part of the paper, the role of the proposed technique is not immediately intuitive (e.g., what constitutes a desired termination and how it differs from undesired outcomes). In particular, Figure 2E is the key illustration that visually demonstrates this difference, but it is not explained at all in the main text. I recommend moving Figure 2E to Figure 1 (right side of the original Fig 1) and introducing it in the Introduction or early in the Results section to clarify the role of the proposed method.

2. (Figures 6–8) These figures show only the actual plasma trajectories, but not the RL-designed or the manually programmed (planned) trajectories. However, the RL-designed trajectory, the manually programmed trajectory in the PCS, and the actual plasma response may all differ. For example, the authors state in Figure 6 that the trajectory was optimized to reduce the LFS gap to prevent VDEs, but from the figure alone, it is unclear whether this was truly suggested by the RL agent, manually reflected in PCS programming, or just occurred randomly regardless of the programmed trajectory. This distinction is crucial for interpreting and evaluating the value of the research. I recommend that the RL-designed trajectory and the PCS-programmed trajectory be also added to Figures 6 and 7 (e.g., using light-colored or dashed lines). In Figure 8, although multiple RL trajectories under uncertainty are shown as black lines, the actual programmed trajectory used in the experiment is not shown. While the caption mentions that the average trajectory was used, it would be helpful to indicate this explicitly in the figure as well.

3. (Page 24) Unlike general RL settings, the RL agent in this study observes only time (Equation 12). This implies that once trained, the agent will always produce the same trajectory, regardless of the initial condition or change of plasma states. Thus, the first question is: when computing the trajectory to input into TCV as shown in Figure 8, how was random uncertainty introduced (like just normal random added to the given trajectory)? Second, if the initial conditions or plasma current differ, does the RL agent need to be re-trained from scratch each time?

Minor Comments:

4. (Page 2) In the Introduction, the authors emphasize that this study focuses on feedforward trajectory design before the

experiment, in contrast to previous RL-based feedback control studies. However, some previous works have also used deep learning-based predictive models in combination with RL to design operation trajectories in a feedforward manner (e.g., [J Seo et al., Nucl. Fusion 61 106010 (2021) & J Seo et al., Nucl. Fusion 62 086049 (2022)]). Although these earlier works may be more limited and less robust to uncertainty compared to the current study, they are similar in concept and deserve to be mentioned as related study.

5. (Figure 2A and others) In Figure 2A, the markers in the legend are difficult to distinguish from those in the graph itself. I suggest modifying the legend (like using boxed legends or other formatting) for easier visual distinction, including in other similar figures.

6. (Figure 2D) The background colors in this graph are not intuitive. At first glance, one might assume that green indicates successful terminations and red indicates failures. However, shot 81635, which was successfully terminated, is marked red, while shot 81741, which had a legacy software issue, is shown in green. Additionally, the x-axis does not follow a monotonic shot order (e.g., 81635 appears after 81741), and there is no explanation for the ambiguous non-red/non-green background colors. It would also be helpful to indicate the threshold values for successful termination (W_{tot} and I_p criteria) directly on the graph. Similar color ambiguity is present in Figure 3 (e.g., mixed green and blue in W_{tot} panel), but without explanation.

7. (Page 6) The authors aim to design trajectories that avoid user-specific limits associated with disruptions and so create models that predict the evolution of relevant quantities. For the readers to understand better, it would be helpful to explicitly define the key quantities correlated with disruptions and the criteria or limits that typically lead to disruptions. Although Table 3 later in the paper provides some indirect answers through the structure of the reward function, it would be better to introduce and explain these disruption criteria earlier in the manuscript. Fusion experts may already be familiar with them, but for a multidisciplinary audience like that of Nature Communications, such clarifications are important.

8. (Tables 1 and 2) Tables 1 and 2 both define input and output variables for the deep learning models. However, in Table 1 (for NSSM), the inputs (actions) are at the bottom and outputs (observations) are at the top, while Table 2 has inputs at the top and outputs at the bottom. This inconsistency may confuse readers, especially since in typical RL settings, observations are inputs and actions are outputs. Unless there is a reason for this ordering, I recommend standardizing the layout for clarity.

9. (Page 6 and Figures 5 & 9) The profile predictor appears to reconstruct 1D profiles from already predicted 0D parameters, rather than predicting the dynamics of the profiles directly. This implies that the information in the 1D profiles is already embedded in the 0D parameters. Therefore, using the reconstructed 1D profiles in NSSM or RL trajectory design might be unnecessary and only increase input complexity. In fact, as far as I understand, the profile predictor does not play a major role in the key process of this study. I'm wondering the role of this profile predictor regarding this study. If it is not essential to the main pipeline, I suggest removing it from the current paper and including it in a follow-up study.

10. (Figure 8) The black lines represent RL trajectories under uncertainty in the actuators. However, in practice, uncertainty (or noise) affects not only the level of the actions or trajectory but also introduces fluctuations in the actuators—like the red or magenta lines in the figure. However, the RL trajectories for a_{minor} remain flat except for level differences. Doesn't fluctuating action noise in real experiments affect the robustness of the designed RL trajectories?

(Remarks on code availability)

Version 1:

Reviewer comments:

Reviewer #1

(Remarks to the Author)

1) In the newly added text at page 4 the authors claim:

“At present, a comprehensive understanding of disruptive limits remains an open problem, motivating many works on machine-learning based prediction of disruptions 37–40.”

Despite being valid, this statement is supported by many self-citing references. Further/different works could be referenced since ML-based prediction of disruptions is a well developed line of research in fusion.

2) Despite the statement of the authors, I consider the role of the profile predictor is still marginal in the description of the work, and recommend showing the plots discussing the 1D profile reconstruction either in the supplementary data or in a new work. On the other hand, for the purposes of the work, I find more interesting the discussion on Figures 16 and 20, now in the supplementary data. These figures allow to discuss the validity of the NSSM for the current application. For instance, in the two successful pulses of Figure 16, the y_{gr} is close to the same limit value as in the one of Figure 20. How is the distinction between these two sets of pulses made?

(Remarks on code availability)

Reviewer #2

(Remarks to the Author)

Thanks for the authors' sincere replies to my previous comments. The authors appropriately answered my previous questions and revised the paper accordingly. Now, I recommend the paper for publication as it is.

(Remarks on code availability)

Responses to Reviewers

Reviewer 1

Thank you for taking the time to provide helpful feedback. We concur that it would be beneficial to increase the statistics of the result; however, it is not straightforward to simply request further shots from an experimental fusion device, especially for this high performance scenario which is highly unusual for TCV.

We have, however, performed additional analyses to increase confidence in the results. This includes running a disruptive shots action trajectory in the RL environment, as you recommended. This also involved a further physics-based analysis confirming that the new RL design trajectories have more vertical stability in the presence of control errors.

We have also made changes to the introductory figures in the paper, in large part in response to feedback from your co-reviewer. We hope the new format makes the results more accessible to a broad audience.

Thank you for pointing out issues with the codebase. It appears we did not archive the key submodule; this issue has been addressed.

Further details are provided in the line item responses below.

Regards,
Allen M. Wang
On Behalf of Coauthors

Line-Item Response

Comment	Response
The work is interesting but due to the scarcity of experiments, the successfully designed rampdowns are only 5, out of a total of 9 optimized ones. For this reason, a further analysis and replication of the experiments is recommended for making robust conclusions.	We would like to point out that: 1) 6 rampdowns were successful 2) Two shots were utilized explicitly to debug an unrelated issue caused by a legacy $1/I_p^2$ term in the PCS, leading to poor radial control during rampdowns. The fact that this issue existed and was finally fixed after the debug shots is shown in Figure 14 in the Supplementary Materials. We

	have chosen to report these shots nevertheless for transparency, and to raise awareness on the often subtle, but important, issues involved in fusion experiments which are often omitted from publications. 3) We performed statistical analysis both with and without these two shots included in the test set. Even with these two shots included in the test set, the results are encouraging.
The article is quite dense, and the authors explain that one model (the NSSM) serves as the training environment for the RL model, however another neural model to predict the temperature and density profiles is mentioned as part of the NSSM. From section 2.2: “The NSSM was initially developed with a neural network predictor for the kinetic profiles on the full ρ grid, and initial training runs found that the profile predictor can accurately predict kinetic profiles given the set of 0D scalars listed in Table 2. Figure 5 provides an example comparison of predictions of the T_e and n_e profiles against Thomson measurements for a full shot in the validation dataset, showing accurate prediction across all phases of the shot. This result corroborates previous findings at NSTX-U that neural networks can accurately predict kinetic profiles given a set of similar 0D scalars³⁴. Given this result suggests most of the relevant profile information is implicitly captured by 0D scalars, the profile predictor was disabled prior to running experiments to accelerate training, hence reported predictions of kinetic profiles are not predict-first. A noteworthy feature of this profile predictor that should be explored in future work is its	We understand why removing the content on the profile predictor would help streamline the paper. However, we want to highlight the fact that the results of our work motivate further research investment in combining established 0D transport simulation methods with neural networks. We believe this is a worthwhile message to preserve, given the large number of ongoing efforts towards developing principles-based transport simulators for control. This message is highlighted in the Discussion: “The ability for a neural network to predict kinetic profiles using 0D scalars, demonstrated both in this work and in prior work⁴³, suggests a data-driven approach may be sufficient for certain control tasks without principles-based transport simulation, which can be extremely computationally expensive and require strong assumptions on edge temperature and density.” Changes Made: added a sentence earlier in the paper (Section 2.2) for clarity of message: “This result also suggests a structured data-driven approach to modeling tokamak transport merits further research, in parallel with several ongoing principles-based efforts.”

ability to function as a Thomson up-sampler, as the input variables are all sampled at a higher time resolution, 1ms, than the TCV Thomson Scattering system, which takes measurements every 17ms.” If the predictor has not been adopted, I suggest removing the parts connected with the profile predictor (e.g section 4.1.3, Figures 5, 9) to avoid confusion, otherwise I would suggest indicating for how many discharges the predictor has been used to train the RL method and if there was any benefit from this approach.	
How were the discharges to train the NSSM model selected? What are the variations in terms of the (observation, action) space?	We provide details in the subsection “Training Data Distribution” in the methods. To provide further clarity, we used the DEFUSE system to gather recent shots that simultaneously have a partial rampdown and sufficient diagnostic availability. A plot of the data distribution in beta, Ip and greenwald fraction space is also shown in that subsection. Changes Made: modified “Training Data Distribution” to clarify that we gathered the most recent shots with sufficient diagnostic availability.
Connected to the previous point, it seems that the vertical instability growth rate (γ_{vgr}) reconstruction by the NSSM is very poor, even with a very low prediction Horizon (Figure 4). Since this parameter is connected to vertical plasma stability, this would explain the VDE event in shot #81751. How large is the uncertainty assumed for the γ_{vgr}?	We discuss the prediction accuracy of γ_{vgr} in Page 6 paragraph two: “The percent errors for γ_{vgr} can be relatively large, but, as shown in Figure 16 this is largely attributable to the small value of γ_{vgr} of limited plasmas as the absolute error is relatively low.” The wide distribution you are seeing is due to the fact that γ_{vgr} on TCV is highly sensitive to control errors, which is discussed in page 11. The uncertainty distributions added to the γ_{vgr} and the variables that are input into the model predicting it were quantified from model prediction error metrics, as discussed in page 23: “In some cases, uncertainty distributions could easily be

	quantified from past experimental data (such as tracking error in the plasma current), or from model prediction accuracy (such as γ_{vgr}), ”
An interesting point could be made by running the control discharges in the NSSM and showing that they were close to disruptive boundaries/expected to disrupt. This would show, despite the low number of experiments, that the NSSM can well represent the system.	We performed this analysis, which shows that the training environment expects a large vertical instability growth rate using the action trajectories from 81101, one of the baseline shots. Given the length and scope of the main paper as it is, and the size of this plot, we believe this result belongs in the supplementary materials. Changes Made: added a figure to the supplementary materials showing NSSM predictions given the action trajectory from a disruptive shot.
Is it possible to interpret the strategies adopted by the RL model by showing which parameters are kept far from critical boundaries?	The right hand side of Figure 8 is meant to illustrate this.
Some other works related to the use of RL in fusion are missing, as applications for ITER [1] and EAST [2].	Thank you for pointing out these works. Changes Made: included references to these two papers in the introduction.
Please add some details on the model adopted (number of layers, neurons) and on how they have been selected/optimized.	Thank you for pointing out this omission. Changes Made: Introduced “Training Methods” to the NSSM section. Added sentences added to “RL Methods” specifying model architecture.
I tried running the code and installing all the dependencies using poetry, but I could not manage to make it work. Most of the python notebooks seem to reproduce the figure and the main results, I did not find the scripts for training the models.	Apologies, it appears that we didn’t archive the submodule containing the dynamics model. We have taken care to make sure it is available in the updated .tar.gz file and poetry install works if the user is using python3.10/11. While we have taken care to make sure notebooks under “paper_plot_generation” are reproducible, execution of the full repo is not possible due to dependencies on large datasets that are under Eurofusion data access restrictions, and thus cannot be made

	public for the foreseeable future. The NSSM training main script is under: “submodules/contrax/contrax/examples/plasma/tcv/train_model.py” RL training scripts are under: “scripts/train_ppo.py” “scripts/train_es.py” Changes Made: updated .tar.gz containing codebase + data files.
--	--

Reviewer 2

Thank you for taking your time to provide helpful feedback in improving the quality of the manuscript, and informing us of important prior works on RL for trajectory optimization in fusion we were not previously aware of. We have adopted your recommendations, which has helped revamp the early presentation of the paper to be more accessible to the reader.

In response to both you and your co-reviewer, we have also performed a further physics-based analysis which corroborates the improved robustness of RL-designed trajectories to vertical instability by introducing control errors into the shot preparation equilibrium solver and solving for the resulting gamma_vgr distributions.

Further details are provided in the line item responses below.

Regards,
Allen M. Wang
On Behalf of Coauthors

Line-Item Response

Comment	Response
1. While the significance and performance of the study are commendable, the storyline in the figures and descriptions is not very clear. Since Nature Communications is a multidisciplinary journal, the content should be understandable even to readers who are not fusion experts. From a non-expert’s	Thank you for this recommendation. We concur that moving Figure 2E to Figure 1 on the right side is helpful, and we have adopted this change. We have also updated the original Figure 1 to a shot that disrupts at high Greenwald Fraction for further clarity. Changes Made: Figure 2E is now part of Figure 1, and Figure 1 updated to a shot that

perspective, when reading the early part of the paper, the role of the proposed technique is not immediately intuitive (e.g., what constitutes a desired termination and how it differs from undesired outcomes). In particular, Figure 2E is the key illustration that visually demonstrates this difference, but it is not explained at all in the main text. I recommend moving Figure 2E to Figure 1 (right side of the original Fig 1) and introducing it in the Introduction or early in the Results section to clarify the role of the proposed method.	more clearly illustrated the issue at hand.
2. (Figures 6–8) These figures show only the actual plasma trajectories, but not the RL-designed or the manually programmed (planned) trajectories. However, the RL-designed trajectory, the manually programmed trajectory in the PCS, and the actual plasma response may all differ. For example, the authors state in Figure 6 that the trajectory was optimized to reduce the LFS gap to prevent VDEs, but from the figure alone, it is unclear whether this was truly suggested by the RL agent, manually reflected in PCS programming, or just occurred randomly regardless of the programmed trajectory. This distinction is crucial for interpreting and evaluating the value of the research. I recommend that the RL-designed trajectory and the PCS-programmed trajectory be also added to Figures 6 and 7 (e.g., using light-colored or dashed lines). In Figure 8, although multiple RL trajectories under uncertainty are shown as black lines, the actual programmed trajectory used in the experiment is not shown. While the caption mentions that the	Thank you for your feedback. We will respond by figure. Changes Made: Figure 6: thank you for your recommendation. We have added the RL-designed PCS-programmed trajectory to this figure. To further confirm that the RL-designed trajectory after adding control error is indeed more robust to the vertical instability, we have performed a further physics-based robustness analysis by introducing minor radius control errors to the RL designed trajectories, solving for equilibria using the free-boundary equilibrium solver used for TCV shot preparation, and then calculating γ_{vgr} distributions for the resulting equilibria. The results (shown in the new Figure 7) show that indeed 82875 is more robust than 81751. Figure 7 (now Figure 8): it is more difficult to show the RL designed trajectories in conjunction with the actual trajectories for this figure, given that the trajectories changed from shot to shot. The purpose of the figure is to show that we progressed over the course of the experiment. However, to provide the information you suggested, we have included the equivalent of Figure 8 for the final two 140kA shots as a part of the supplementary information.

average trajectory was used, it would be helpful to indicate this explicitly in the figure as well.	Figure 8 (now Figure 9): this is a good suggestion, we have included the mean of the action trajectories as a solid thick black line as well, and included a comment on this in the caption.
3. (Page 24) Unlike general RL settings, the RL agent in this study observes only time (Equation 12). This implies that once trained, the agent will always produce the same trajectory, regardless of the initial condition or change of plasma states. Thus, the first question is: when computing the trajectory to input into TCV as shown in Figure 8, how was random uncertainty introduced (like just normal random added to the given trajectory)? Second, if the initial conditions or plasma current differ, does the RL agent need to be re-trained from scratch each time?	This is a good point. The uncertainty distribution for initial conditions specified is shown in Section 4.4: Uncertainty Model. The idea is to have a single action trajectory that is likely to succeed across a range of initial conditions. If the initial conditions change by more than anticipated, then indeed retraining needs to happen (as was the case for 140kA vs. 170kA). Changes Made: we have introduced a sentence in 4.5 to make this more clear: “Given that time is the only observable, but there exists different physical conditions in the parallel training environments which are unobservable to the policy, the reward maximization process yields a trajectory that is designed to succeed across the different conditions specified in Subsection \ref{subsec:uncertainty_model}.”
4. (Page 2) In the Introduction, the authors emphasize that this study focuses on feedforward trajectory design before the experiment, in contrast to previous RL-based feedback control studies. However, some previous works have also used deep learning-based predictive models in combination with RL to design operation trajectories in a feedforward manner (e.g., [J Seo et al., Nucl. Fusion 61 106010 (2021) & J Seo et al., Nucl. Fusion 62 086049 (2022)]). Although these earlier works may be more limited and less robust to uncertainty compared to the current	Thank you for informing us of this work. We have included references. Changes Made: we have included both citations and introduced the papers in the introduction: “A similar approach has previously been demonstrated at KSTAR for designing feed-forward trajectories that reach target states \cite{seo2021feedforward,seo2022development}.”

study, they are similar in concept and deserve to be mentioned as related study.	
5. (Figure 2A and others) In Figure 2A, the markers in the legend are difficult to distinguish from those in the graph itself. I suggest modifying the legend (like using boxed legends or other formatting) for easier visual distinction, including in other similar figures.	Thank you for pointing out this deficiency. Changes Made: added a box to the legend.
6. (Figure 2D) The background colors in this graph are not intuitive. At first glance, one might assume that green indicates successful terminations and red indicates failures. However, shot 81635, which was successfully terminated, is marked red, while shot 81741, which had a legacy software issue, is shown in green. Additionally, the x-axis does not follow a monotonic shot order (e.g., 81635 appears after 81741), and there is no explanation for the ambiguous non-red/non-green background colors. It would also be helpful to indicate the threshold values for successful termination (W_{tot} and I_p criteria) directly on the graph. Similar color ambiguity is present in Figure 3 (e.g., mixed green and blue in W_{tot} panel), but without explanation.	Thank you for pointing out the non-monotonic ordering. This was somewhat embarrassing given that it was 81741 and 81745 that were the debug shots. We have addressed this issue. Your feedback has helped us recognize there is a better way to communicate our results. We have now replaced the histogram with a scatter plot in (W_{tot} and I_p) space showing the shots and a table of statistics embedded in the plot. We realized it likely makes more sense to have this plot in the “Overview” as opposed to having the shot-by-shot breakdown in the overview. The shot-by-shot breakdown is now its own separate figure . We have added the goal I_p to the scatter plot. As we note in a new footnote, we initially chose a goal stored energy, 0.5kJ, that proved unrealistic to diagnose as equilibrium reconstruction no longer converges reliably at such low plasma current and stored energy. Thus, we would like to primarily focus on reaching the goal I_p as it can be diagnosed relatively reliably without reconstruction. Changes Made: 81635 + 81741 are now in their proper order. Figure 3 is now a scatter plot in W_{tot} and I_p space and is part of the overview. The shot-by-shot breakdown is now a separate figure. Added footnote on how we chose an overtly optimistically low energy that can be reached.

7. (Page 6) The authors aim to design trajectories that avoid user-specific limits associated with disruptions and so create models that predict the evolution of relevant quantities. For the readers to understand better, it would be helpful to explicitly define the key quantities correlated with disruptions and the criteria or limits that typically lead to disruptions. Although Table 3 later in the paper provides some indirect answers through the structure of the reward function, it would be better to introduce and explain these disruption criteria earlier in the manuscript. Fusion experts may already be familiar with them, but for a multidisciplinary audience like that of Nature Communications, such clarifications are important.	Thank you for the feedback. We agree an earlier discussion would be beneficial. Changes Made: We have amended the end of paragraph 1 in 2.1 to introduce the issue of disruptive limits and the constraints we chose to consider in this work.
8. (Tables 1 and 2) Tables 1 and 2 both define input and output variables for the deep learning models. However, in Table 1 (for NSSM), the inputs (actions) are at the bottom and outputs (observations) are at the top, while Table 2 has inputs at the top and outputs at the bottom. This inconsistency may confuse readers, especially since in typical RL settings, observations are inputs and actions are outputs. Unless there is a reason for this ordering, I recommend standardizing the layout for clarity.	Thank you for the suggestion. Changes Made: We have edited Table 1 to include actions on top and observations on the bottom.
9. (Page 6 and Figures 5 & 9) The profile predictor appears to reconstruct 1D profiles from already predicted 0D parameters, rather than predicting the dynamics of the profiles directly. This implies that the information in the 1D profiles is already	You are correct. One of the findings of this study is that details of 1D kinetic profiles did not appear significant to achieve this result. Even though the 1D profile predictor is not necessary to obtain the core result, we believe its inclusion in the paper contributes to ongoing dialogue about the priority of developing high-fidelity transport simulators

embedded in the 0D parameters. Therefore, using the reconstructed 1D profiles in NSSM or RL trajectory design might be unnecessary and only increase input complexity. In fact, as far as I understand, the profile predictor does not play a major role in the key process of this study. I'm wondering the role of this profile predictor regarding this study. If it is not essential to the main pipeline, I suggest removing it from the current paper and including it in a follow-up study.

for plasma control tasks, given the large volume of effort currently expended in that direction [1, 2, 3]. As we discuss in the Discussion section:

“The ability for a neural network to predict kinetic profiles using 0D scalars, demonstrated both in this work and in prior work [36], suggests a data-driven approach may be sufficient for certain control tasks without principles-based transport simulation, which can be extremely computationally expensive and require strong assumptions on edge temperature and density.”

However, your comment helped us realize it will be worthwhile to make this message clear earlier in the paper as well.

Changes Made: added a sentence earlier in the paper (Section 2.2) for clarity of message: “This result also suggests a structured data-driven approach to modeling tokamak transport merits further research, in parallel with several ongoing principles-based efforts.”

[1] Citrin, Jonathan, et al. "Torax: A fast and differentiable tokamak transport simulator in jax." arXiv preprint arXiv:2406.06718 (2024).

[2] Muraca, Marco, et al. "Reduced transport models for a tokamak flight simulator." Plasma Physics and Controlled Fusion 65.3 (2023): 035007.

[3] Meneghini, O., et al. "FUSE (Fusion Synthesis Engine): A Next Generation Framework for Integrated Design of Fusion Pilot Plants." arXiv preprint arXiv:2409.05894 (2024).

10. (Figure 8) The black lines represent RL trajectories under uncertainty in the actuations. However, in practice, uncertainty (or noise) affects not only the level of the actions or trajectory but also introduces fluctuations in the actuations—like the red or magenta lines in the figure. However, the RL trajectories for a_{minor} , remain flat except for level

You are correct. This is a limitation of our uncertainty model.

Changes Made: Included the following sentence at the end of the “Uncertainty Model” subsection:

“In addition, the uncertainty model employed does not account for time-varying fluctuations in uncertain variables; future work should employ time-varying stochastic processes.

differences. Doesn't fluctuating action noise in real experiments affect the robustness of the designed RL trajectories?	Both of these limitations further highlight the need to advance experimental uncertainty quantification and robust control in the context of fusion plasma control.”
--	--

Reviews

Reviewer #1 (Remarks to the Author):

The work revolves around the use of reinforcement learning (RL) for the optimization of the ramp-down trajectory at TCV, an experimental device in Switzerland. The idea of RL is that the model can learn how to interact with the system by trial and error and requires a sufficiently accurate model of the system. In this case, also the system is approximated by a Neural State Space Model (NSSM), that is a Neural Network which can approximate the state space representation mixing a data driven loss and numerical integration of the (net+system) dynamics.

The work is interesting but due to the scarcity of experiments, the successfully designed rampdowns are only 5, out of a total of 9 optimized ones. For this reason, a further analysis and replication of the experiments is recommended for making robust conclusions.

Moreover, I have the following comments on the work:

NSSM and training environment for the RL system

The article is quite dense, and the authors explain that one model (the NSSM) serves as the training environment for the RL model, however another neural model to predict the temperature and density profiles is mentioned as part of the NSSM. From section 2.2: “The NSSM was initially developed with a neural network predictor for the kinetic profiles on the full ρ grid, and initial training runs found that the profile predictor can accurately predict kinetic profiles given the set of 0D scalars listed in Table 2.

Figure 5 provides an example comparison of predictions of the T_e and n_e profiles against Thomson measurements for a full shot in the validation dataset, showing accurate prediction across all phases of the shot. This result corroborates previous findings at NSTX-U that neural networks can accurately predict kinetic profiles given a set of similar 0D scalars³⁴. Given this result suggests most of the relevant profile information is implicitly captured by 0D scalars, the profile predictor was disabled prior to running experiments to accelerate training, hence reported predictions of kinetic profiles are not predict-first. A noteworthy feature of this profile predictor that should be explored in future work is its ability to function as a Thomson up-sampler, as the input variables are all sampled at a

higher time resolution, 1ms, than the TCV Thomson Scattering system, which takes measurements every 17ms.”

If the predictor has not been adopted, I suggest removing the parts connected with the profile predictor (e.g section 4.1.3, Figures 5, 9) to avoid confusion, otherwise I would suggest indicating for how many discharges the predictor has been used to train the RL method and if there was any benefit from this approach.

How were the discharges to train the NSSM model selected? What are the variations in terms of the (observation, action) space?

Connected to the previous point, it seems that the vertical instability growth rate (γ_{vgr}) reconstruction by the NSSM is very poor, even with a very low prediction Horizon (Figure 4). Since this parameter is connected to vertical plasma stability, this would explain the VDE event in shot #81751. How large is the uncertainty assumed for the γ_{vgr} ?

An interesting point could be made by running the control discharges in the NSSM and showing that they were close to disruptive boundaries/expected to disrupt. This would show, despite the low number of experiments, that the NSSM can well represent the system.

Is it possible to interpret the strategies adopted by the RL model by showing which parameters are kept far from critical boundaries?

Other comments

Some other works related to the use of RL in fusion are missing, as applications for ITER [1] and EAST [2].

Please add some details on the model adopted (number of layers, neurons) and on how they have been selected/optimized.

I tried running the code and installing all the dependencies using poetry, but I could not manage to make it work. Most of the python notebooks seem to reproduce the figure and the main results, I did not find the scripts for training the models.

[1] S. Dubbioso, G. De Tommasi, A. Mele, G. Tartaglione, M. Ariola, and A. Pironti, ‘A Deep Reinforcement Learning approach for Vertical Stabilization of tokamak plasmas’, Fusion Engineering and Design, vol. 194, p. 113725, Sep. 2023, doi: 10.1016/j.fusengdes.2023.113725.

[2] G. De Tommasi, S. Dubbioso, Y. Huang, Z. P. Luo, A. Mele, and B. J. Xiao, ‘A RL-based Vertical Stabilization System for the EAST tokamak’, in 2022 American Control Conference (ACC), Jun. 2022, pp. 5328–5333. doi: 10.23919/ACC53348.2022.9867499.

Reviewer #1 (Remarks on code availability):

Partial review of the code due to the difficulties in installing all the proper libraries
The readme file instructs on using poetry, but after finding the correct version of poetry I ran into an error:
Directory TokaGym\submodules\contrax for contrax does not seem to be a Python package

Reviewer #2 (Remarks to the Author):

General Comments:

This study leverages deep learning (NSSM) to predict plasma ramp-down dynamics in the TCV tokamak and applies reinforcement learning (RL) to design an optimal trajectory for stable termination during this phase. Notably, ramp-down in fusion reactors is physically and technically prone to instability, often associated with plasma disruptions. Thus, ensuring stable control and termination in this phase is critical for future large-scale devices such as ITER. This study combines a predictive model and RL on TCV to design a stable plasma termination trajectory and demonstrates robust outcomes through incremental re-training with actual experiments. This represents a significant milestone in fusion research. I believe that, provided the following comments are adequately addressed, the manuscript could be suitable for publication in Nature Communications.

Major Comments:

1. While the significance and performance of the study are commendable, the storyline in the figures and descriptions is not very clear. Since Nature Communications is a multidisciplinary journal, the content should be understandable even to readers who are not fusion experts. From a non-expert's perspective, when reading the early part of the paper, the role of the proposed technique is not immediately intuitive (e.g., what constitutes a desired termination and how it differs from undesired outcomes). In particular, Figure 2E is the key illustration that visually demonstrates this difference, but it is not explained at all in the main text. I recommend moving Figure 2E to Figure 1 (right side of the original Fig 1) and introducing it in the Introduction or early in the Results section to clarify the role of the proposed method.

2. (Figures 6–8) These figures show only the actual plasma trajectories, but not the RL-designed or the manually programmed (planned) trajectories. However, the RL-designed trajectory, the manually programmed trajectory in the PCS, and the actual plasma response may all differ. For example, the authors state in Figure 6 that the trajectory was optimized to reduce the LFS gap to prevent VDEs, but from the figure alone, it is unclear whether this was truly suggested by the RL agent, manually reflected in PCS programming, or just occurred randomly regardless of the programmed trajectory. This distinction is crucial for interpreting and evaluating the value of the research. I recommend that the RL-designed trajectory and the PCS-programmed trajectory be also added to Figures 6 and 7 (e.g., using light-colored or dashed lines).

In Figure 8, although multiple RL trajectories under uncertainty are shown as black lines, the actual programmed trajectory used in the experiment is not shown. While the caption mentions that the average trajectory was used, it would be helpful to indicate this explicitly in the figure as well.

3. (Page 24) Unlike general RL settings, the RL agent in this study observes only time (Equation 12). This implies that once trained, the agent will always produce the same trajectory, regardless of the initial condition or change of plasma states. Thus, the first question is: when computing the trajectory to input into TCV as shown in Figure 8, how was random uncertainty introduced (like just normal random added to the given trajectory)? Second, if the initial conditions or plasma current differ, does the RL agent need to be re-trained from scratch each time?

Minor Comments:

4. (Page 2) In the Introduction, the authors emphasize that this study focuses on feedforward trajectory design before the experiment, in contrast to previous RL-based feedback control studies. However, some previous works have also used deep learning-based predictive models in combination with RL to design operation trajectories in a feedforward manner (e.g., [J Seo et al., Nucl. Fusion 61 106010 (2021) & J Seo et al., Nucl. Fusion 62 086049 (2022)]). Although these earlier works may be more limited and less robust to uncertainty compared to the current study, they are similar in concept and deserve to be mentioned as related study.

5. (Figure 2A and others) In Figure 2A, the markers in the legend are difficult to distinguish from those in the graph itself. I suggest modifying the legend (like using boxed legends or other formatting) for easier visual distinction, including in other similar figures.

6. (Figure 2D) The background colors in this graph are not intuitive. At first glance, one might assume that green indicates successful terminations and red indicates failures.

However, shot 81635, which was successfully terminated, is marked red, while shot 81741, which had a legacy software issue, is shown in green. Additionally, the x-axis does not follow a monotonic shot order (e.g., 81635 appears after 81741), and there is no explanation for the ambiguous non-red/non-green background colors. It would also be helpful to indicate the threshold values for successful termination (W_{tot} and I_p criteria) directly on the graph. Similar color ambiguity is present in Figure 3 (e.g., mixed green and blue in W_{tot} panel), but without explanation.

7. (Page 6) The authors aim to design trajectories that avoid user-specific limits associated with disruptions and so create models that predict the evolution of relevant quantities. For the readers to understand better, it would be helpful to explicitly define the key quantities correlated with disruptions and the criteria or limits that typically lead to disruptions. Although Table 3 later in the paper provides some indirect answers through the structure of the reward function, it would be better to introduce and explain these disruption criteria earlier in the manuscript. Fusion experts may already be familiar with them, but for a multidisciplinary audience like that of Nature Communications, such clarifications are important.

8. (Tables 1 and 2) Tables 1 and 2 both define input and output variables for the deep learning models. However, in Table 1 (for NSSM), the inputs (actions) are at the bottom and outputs (observations) are at the top, while Table 2 has inputs at the top and outputs at the bottom. This inconsistency may confuse readers, especially since in typical RL settings, observations are inputs and actions are outputs. Unless there is a reason for this ordering, I recommend standardizing the layout for clarity.

9. (Page 6 and Figures 5 & 9) The profile predictor appears to reconstruct 1D profiles from already predicted 0D parameters, rather than predicting the dynamics of the profiles directly. This implies that the information in the 1D profiles is already embedded in the 0D parameters. Therefore, using the reconstructed 1D profiles in NSSM or RL trajectory design might be unnecessary and only increase input complexity. In fact, as far as I understand, the profile predictor does not play a major role in the key process of this study. I'm wondering the role of this profile predictor regarding this study. If it is not essential to the main pipeline, I suggest removing it from the current paper and including it in a follow-up study.

10. (Figure 8) The black lines represent RL trajectories under uncertainty in the actuations. However, in practice, uncertainty (or noise) affects not only the level of the actions or trajectory but also introduces fluctuations in the actuations—like the red or magenta lines in the figure. However, the RL trajectories for a_{minor} remain flat except for level differences. Doesn't fluctuating action noise in real experiments affect the robustness of the designed RL trajectories?

Reviewer Comment: 1) In the newly added text at page 4 the authors claim: "At present, a comprehensive understanding of disruptive limits remains an open problem, motivating many works on machine-learning based prediction of disruptions 37–40."

Despite being valid, this statement is supported by many self-citing references. Further/different works could be referenced since ML-based prediction of disruptions is a well developed line of research in fusion.

Response: Thank you for your suggestion. We would first like to clarify that only two out of the four cited works could be fairly characterized as self-citation. 38 is an overview paper for the international project ITER, involving a large portion of the world's experts, which inevitably leads to involvement of authors of this paper as well. 40 does not involve any authors of this paper whatsoever. To further address the balance, we have added a reference to Vega et al., 2022 Nature Physics.

Reviewer Comment: 2) Despite the statement of the authors, I consider the role of the profile predictor is still marginal in the description of the work, and recommend showing the plots discussing the 1D profile reconstruction either in the supplementary data or in a new work. On the other hand, for the purposes of the work, I find more interesting the discussion on Figures 16 and 20, now in the supplementary data. These figures allow to discuss the validity of the NSSM for the current application. For instance, in the two successful pulses of Figure 16, the y_{vgr} is close to the same limit value as in the one of Figure 20. How is the distinction between these two sets of pulses made?

Response: Thank you for your perspective. Regarding the profile prediction content, we would like to re-iterate our opinion that the result is informative for the community given the multiple current research efforts to do this with a principles-based approach [1, 2, 3].

Regarding Figures 16 and 20, we would like to highlight that they are new figures introduced as a part of the revision. We would like to point out that in Figure 20, the distribution travels much further into the soft constraint region than in Figure 16, especially in the milliseconds immediately preceding the disruption.

[1] Teplukhina, A. A., et al. "Simulation of profile evolution from ramp-up to ramp-down and optimization of tokamak plasma termination with the RAPTOR code." *Plasma Physics and Controlled Fusion* 59.12 (2017): 124004.

[2] Citrin, Jonathan, et al. "TORAX: A fast and differentiable tokamak transport simulator in JAX." arXiv preprint arXiv:2406.06718 (2024).

[3] Meneghini, O., et al. "FUSE (Fusion Synthesis Engine): A next generation framework for integrated design of fusion pilot plants." arXiv preprint arXiv:2409.05894 (2024).